# Autophagosome development and chloroplast segmentation occur synchronously for piecemeal degradation of chloroplasts

**Masanori Izumi[1,2]\*, Sakuya Nakamura[2], Kohei Otomo[3,4,5,6,7], Hiroyuki Ishida[8], Jun Hidema[9†], Tomomi Nemoto[3,4,5,6], Shinya Hagihara[2]**

[1]Frontier Research Institute for Interdisciplinary Sciences (FRIS), Tohoku University, Sendai, Japan; [2]Center for Sustainable Resource Science (CSRS), RIKEN, Wako, Japan; [3]Exploratory Research Center on Life and Living Systems (ExCELLs), National Institutes of Natural Sciences, Okazaki, Japan; [4]National Institute for Physiological Sciences, National Institutes of Natural Sciences, Okazaki, Japan; [5]The Graduate University for Advanced Studies, SOKENDAI, Okazaki, Japan; [6]Research Institute for Electronic Science, Hokkaido University, Sapporo, Japan; [7]Graduate School of Medicine, Juntendo University, Tokyo, Japan; [8]Graduate School of Agricultural Science, Tohoku University, Sendai, Japan; [9]Graduate School of Life Sciences, Tohoku University, Sendai, Japan

**\*For correspondence:** masanori.izumi@riken.jp

**Present address:** [†]Institute for Advanced Academic Research, Graduate School of Horticulture Research Center for Space Agriculture and Horticulture, Chiba Univeristy, Matsudo, Japan

**Competing interest:** The authors declare that no competing interests exist.

**Sent for Review** 11 October 2023
**Preprint posted** 14 October 2023
**Reviewed preprint posted** 19 December 2023
**Reviewed preprint revised** 11 September 2024
**Version of Record published** 07 November 2024

## eLife assessment

This manuscript investigates how chloroplasts are broken down during light-limiting conditions as plants reorganize their energy-producing organelles during carbon limitation. The authors provide **compelling** live-cell imaging data of plastids and **solid** quantification of events, documenting that buds form on the surface of chloroplasts and pinch away, then associate with the vacuole via a mechanism that depends on autophagy machinery, but not plastid division machinery. This manuscript provides **valuable** groundwork for other scientists studying the regulation and breakdown of energy-producing organelles, including chloroplasts and mitochondria.

**Abstract** Plants distribute many nutrients to chloroplasts during leaf development and maturation. When leaves senesce or experience sugar starvation, the autophagy machinery degrades chloroplast proteins to facilitate efficient nutrient reuse. Here, we report on the intracellular dynamics of an autophagy pathway responsible for piecemeal degradation of chloroplast components. Through live-cell monitoring of chloroplast morphology, we observed the formation of chloroplast budding structures in sugar-starved leaves. These buds were then released and incorporated into the vacuolar lumen as an autophagic cargo termed a Rubisco-containing body. The budding structures did not accumulate in mutants of core autophagy machinery, suggesting that autophagosome creation is required for forming chloroplast buds. Simultaneous tracking of chloroplast morphology and autophagosome development revealed that the isolation membranes of autophagosomes interact closely with part of the chloroplast surface before forming chloroplast buds. Chloroplasts then protrude at the site associated with the isolation membranes, which divide synchronously with autophagosome maturation. This autophagy-related division does not require DYNAMIN-RELATED PROTEIN 5B, which constitutes the division ring for chloroplast proliferation in growing leaves. An unidentified division machinery may thus fragment chloroplasts for degradation in coordination with the development of the chloroplast-associated isolation membrane.

## Introduction

Organelle morphology changes dynamically in response to fluctuations in cell functions, developmental stages, and environmental cues. Plastids are plant-specific organelles that differentiate into chloroplasts in green tissues to perform photosynthesis. Plastids further serve as hubs for metabolic pathways such as the biosynthesis of amino acids and plant hormones. Dynamic changes in plastid morphology also act as switches initiating biological programs (*Osteryoung and Pyke, 2014*). For instance, chloroplasts in leaf pavement cells form stroma-filled tubular extensions termed stromules to activate programmed cell death as an immune response upon perception of an invading pathogen,

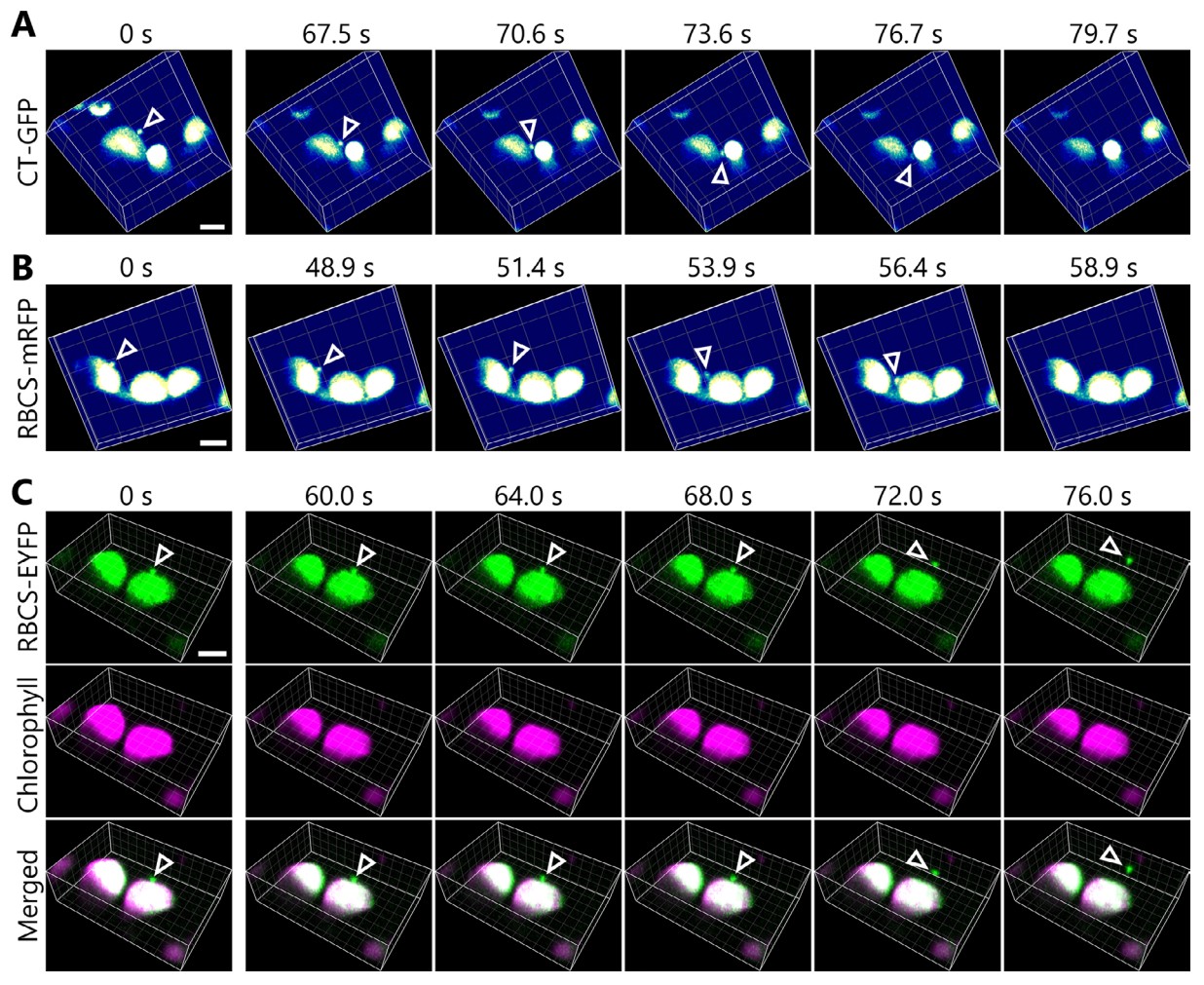

**Figure 1.** Chloroplast buds are released in sugar-starved leaves. Time-lapse observations of three-dimensional (3D) reconstructed chloroplast morphology in *Arabidopsis* mesophyll cells accumulating chloroplast stroma–targeted fluorescent markers. A leaf from a plant accumulating chloroplast stroma–targeted GFP (CT-GFP) (**A**), RBCS-mRFP (**B**), or RBCS-EYFP (**C**) was incubated in sugar-free solution in darkness for 5 hr from dawn (**A**), 21 hr (**B**), or 24 hr from the period in light (**C**) and then observed through a two-photon excitation microscope equipped with a confocal spinning-disk unit. Second rosette leaves from 20- to 22-day-old plants were used. Images in (**A–C**) are still frames from *Videos 1–3*, respectively. Time scales above the images indicate the elapsed time from the start of the respective videos. Arrowheads indicate chloroplast budding structures. Scale bars, 5 μm. In (**C**), green, RBCS-EYFP; magenta, chlorophyll fluorescence. In the merged images, the overlapping regions of RBCS-EYFP and chlorophyll signals appear white.

The online version of this article includes the following source data and figure supplement(s) for figure 1:

**Figure supplement 1.** Accumulation of chloroplast stroma components in the vacuole via autophagy.

**Figure supplement 1—source data 1.** Source data for the graph in *Figure 1—figure supplement 1*.

**Figure supplement 2.** The release of chloroplast buds by autophagy contributes to the decline in chloroplast stroma volume.

**Figure supplement 2—source data 1.** Source data for the graph in *Figure 1—figure supplement 2*.

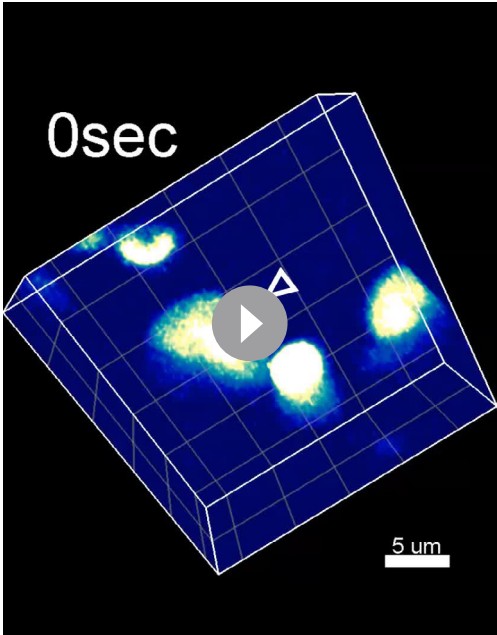

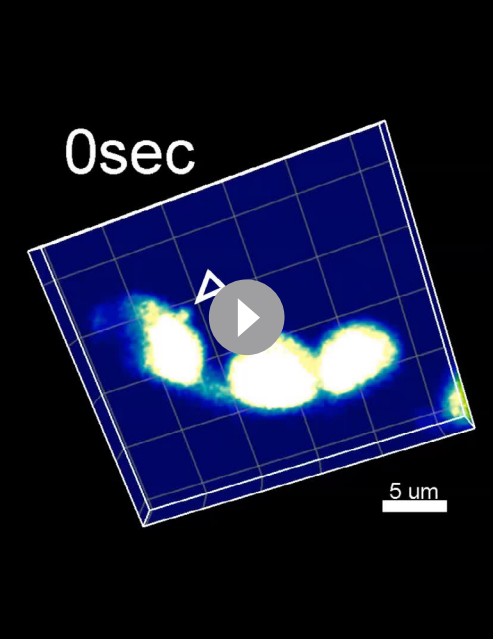

**Video 1.** Release of a chloroplast bud as visualized by chloroplast stroma–targeted GFP. The second rosette leaf from a 21-day-old plant accumulating the chloroplast stroma–targeted GFP (CT-GFP) was incubated in sugar-free solution in darkness for time-lapse imaging with a two-photon excitation microscope equipped with a confocal spinning-disk unit. Arrowhead indicates a chloroplast budding structure. Three-dimensional (3D) reconstructed images (8 μm in depth) acquired about every 3 s are displayed at 10 frames/s. Scale bar, 5 μm. This video was used to generate *Figure 1A*.
https://elifesciences.org/articles/93232/figures#video1

**Video 2.** Release of a chloroplast bud as visualized by RBCS-mRFP. The second rosette leaf from a 22-day-old plant accumulating the chloroplast stroma marker RBCS-mRFP was incubated in sugar-free solution in darkness for time-lapse imaging with a two-photon excitation microscope equipped with a confocal spinning-disk unit. Arrowhead indicates a chloroplast budding structure. Three-dimensional (3D) reconstructed images (2 μm in depth) acquired about every 1.2 s are displayed at 10 frames/s. Scale bar, 5 μm. This video was used to generate *Figure 1B*.
https://elifesciences.org/articles/93232/figures#video2

in a process regulated by the plant hormone salicylic acid (SA) (*Caplan et al., 2015*). Stromules are thin, tubular structures of plastid stroma less than 1 μm in diameter (*Hanson and Sattarzadeh, 2011*).

Leaf aging is closely associated with morphological and functional changes of plastids. At the early stage of leaf development, the undifferentiated form of plastids, termed proplastids, are converted to green chloroplasts as their population expands through active division. In *Arabidopsis* (*Arabidopsis thaliana*), the coordination of cytoplasmic ring structures comprising DYNAMIN-RELATED PROTEIN 5B (DRP5B), also called ACCUMULATION AND REPLICATION OF CHLOROPLASTS 5 (ARC5), and intrachloroplastic ring structures consisting of FILAMENTING TEMPERATURE-SENSITIVE Z (FTSZ) proteins mediates chloroplast division (*Chen et al., 2018*). In mature leaves, spherical chloroplasts occupy almost all of the cytoplasm and manifest high photosynthetic activity. When older leaves later senesce, the size and number of chloroplasts gradually decrease (*Martinoia et al., 1983*; *Mae et al., 1984*). Younger leaves closer to the shoot apex sometimes shade older leaves beneath them, thus perturbing photosynthesis by decreasing the amount of light reaching those leaves and causing local sugar starvation. In such sugar-starved leaves, senescence is accelerated to facilitate the relocation of nutrients into newly developing tissues (*Weaver and Amasino, 2001*; *Ono et al., 2013*; *Law et al., 2018*). Autophagy is an intracellular degradation machinery that contributes to the decline in chloroplast volume and numbers during leaf senescence (*Wada et al., 2009*).

Autophagy is a ubiquitous mechanism that transports cytoplasmic components to the vacuolar/lysosomal compartment for degradation in eukaryotic cells. The major autophagy pathway known as macroautophagy, or simply autophagy, begins with the assembly of a membrane termed the isolation membrane (also known as a phagophore) that forms a double-membrane structure termed the autophagosome and sequesters a portion of the cytoplasm. Core AUTOPHAGY (ATG) proteins (ATG1–10,

12–14, 16, and 18) are highly conserved among plants, yeasts, and mammals (*Yoshimoto and Ohsumi, 2018*; *Nakamura et al., 2021a*) and are required for the initiation and elongation of the isolation membrane (*Nakatogawa, 2020*). The outer membrane of autophagosomes fuses with the vacuolar/lysosomal membrane, resulting in the formation of autophagic bodies whose cargos are digested by vacuolar/lysosomal hydrolases. Our previous studies identified two types of autophagy for chloroplast degradation in mature leaves (*Ishida et al., 2014*; *Nakamura and Izumi, 2018*). Senescence and sugar starvation preferentially activate piecemeal degradation of chloroplast components involving a specific type of autophagosomal cargo termed a Rubisco-containing body (RCB), leading to smaller chloroplasts (*Ishida et al., 2008*; *Wada et al., 2009*; *Izumi et al., 2015*). The second type of autophagy, termed chlorophagy, removes entire unnecessary chloroplasts (*Izumi et al., 2017*; *Nakamura et al., 2018*), thereby modulating the number of chloroplasts in a cell (*Wada et al., 2009*; *Izumi et al., 2017*). Chlorophagy is likely another form of autophagy, termed microautophagy, during which the vacuolar membrane directly participates in sequestering the degradation target in the absence of encapsulation by autophagosomes (*Izumi et al., 2017*; *Lee et al., 2023*).

Mitochondria are another type of energy-producing organelle derived from endosymbiosis. Dynamin-related proteins (DRPs) also mediate mitochondrial division (*Giacomello et al., 2020*); the involvement of DRP-mediated organelle division in mitochondrion-targeted autophagy—termed mitophagy—in mammals and budding yeast (*Saccharomyces cerevisiae*) has been under debate (*Chen et al., 2022*). Drp1 and Dnm1 (Dynamin-related 1) participate in mitochondrial fission in mammals and budding yeast, respectively. Previous studies have suggested that segmented mitochondria resulting from Drp1/Dnm1-mediated fission become a target for degradation by autophagosomes (*Twig et al., 2008*; *Rambold et al., 2011*; *Abeliovich et al., 2013*). Notably, Drp1/Dnm1-independent mitophagy has also been observed (*Yamashita et al., 2016*), whereby mitochondrial division occurs concomitantly with the development of the mitochondrion-associated isolation membrane, forming an autophagosome that specifically contains mitochondrial components

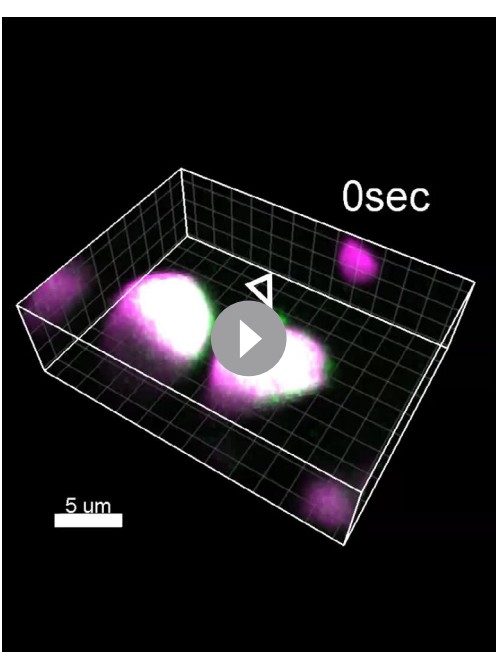

**Video 3.** Tracking of the stroma marker and chlorophyll fluorescence during the release of a chloroplast bud. The second rosette leaf from a 20-day-old plant accumulating the chloroplast stroma marker RBCS-EYFP was incubated in sugar-free solution in darkness for time-lapse imaging with a two-photon excitation microscope equipped with a confocal spinning-disk unit. Arrowhead indicates a chloroplast budding structure. Three-dimensional (3D) reconstructed images (8 μm in depth) acquired every 4 s are displayed at 10 frames/s. Scale bar, 5 μm. Green, RBCS-EYFP; magenta, chlorophyll fluorescence. Only the video of the merged channels is shown. The overlapping regions of RBCS-EYFP and chlorophyll signals appear white. This video was used to generate *Figure 1C*.
https://elifesciences.org/articles/93232/figures#video3

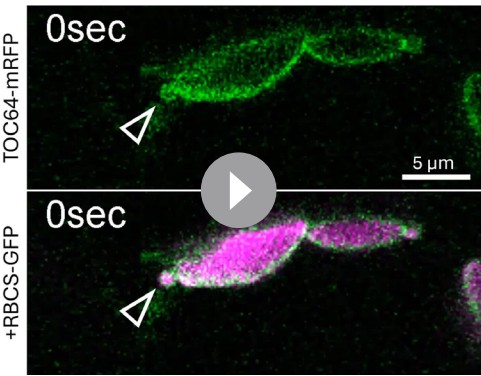

**Video 4.** A released chloroplast bud contains the outer envelope marker TOC64-mRFP. The second rosette leaf from a 21-day-old plant accumulating the chloroplast stroma marker RBCS-GFP and the outer envelope marker TOC64-mRFP was incubated in sugar-free solution in darkness for time-lapse imaging. Arrowheads indicate chloroplast budding structures. Images acquired every 2 s are displayed at 10 frames/s. Scale bar, 5 μm. Green, TOC64-mRFP; magenta, RBCS-GFP. The video contains TOC64-mRFP and the merged channels. This video was used to generate *Figure 2B*.
https://elifesciences.org/articles/93232/figures#video4

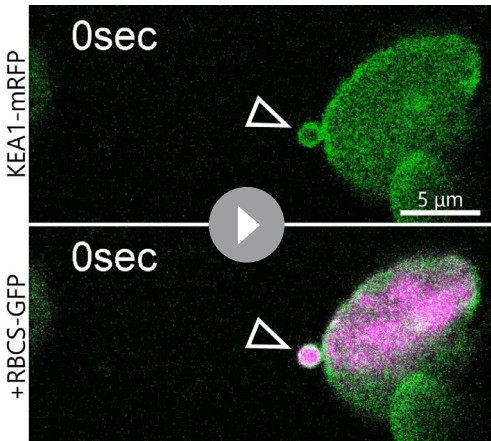

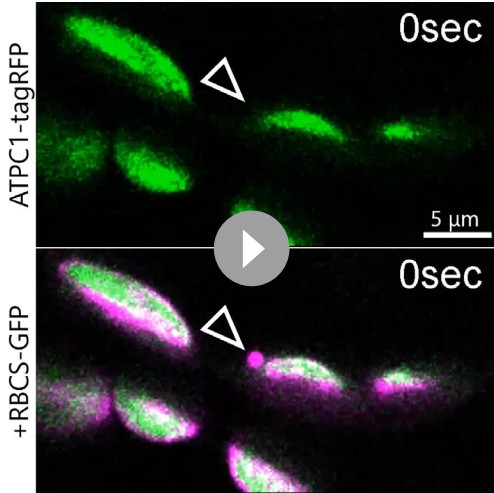

**Video 5.** A released chloroplast bud contains the inner envelope marker KEA1-mRFP. The second rosette leaf from a 22-day-old plant accumulating the chloroplast stroma marker RBCS-GFP and the inner envelope marker KEA1-mRFP was incubated in sugar-free solution in the dark for time-lapse imaging. Arrowheads indicate chloroplast budding structures. Images acquired every 2 s are displayed at 10 frames/s. Scale bar, 5 μm. Green, KEA1-mRFP; magenta, RBCS-GFP. The video contains KEA1-mRFP and the merged channels. This video was used to generate *Figure 2— figure supplement 2*.

https://elifesciences.org/articles/93232/figures#video5

**Video 6.** A released chloroplast bud does not contain the thylakoid membrane marker ATPC1-tagRFP. The second rosette leaf from a 21-day-old plant accumulating the chloroplast stroma marker RBCS-GFP and the thylakoid membrane marker ATPC1-tagRFP was incubated in sugar-free solution in darkness for time-lapse imaging. Arrowheads indicate chloroplast budding structures. Images acquired every 2 s are displayed at 10 frames/s. Scale bar, 5 μm. Green, ATPC1-tagRFP; magenta, RBCS-GFP. The video contains ATPC1-tagRFP and the merged channels. This video was used to generate *Figure 2D*.

https://elifesciences.org/articles/93232/figures#video6

termed the mitophagosome. A recent study identified the mitochondrial intermembrane-space protein mitofissin (also called Atg44), required for mitochondrial fission during yeast mitophagy (*Fukuda et al., 2023*).

Plant autophagic bodies and RCBs are typically around 1 μm in diameter (*Chiba et al., 2003*; *Yoshimoto et al., 2004*), which is smaller than chloroplasts. Thus, piecemeal-type chloroplast autophagy via RCBs must start with division of the chloroplast segments that are to be degraded. However, how a fragment of a chloroplast can be transported to the vacuole remains poorly understood. Here, to characterize the underlying intracellular dynamics, we performed high-resolution imaging analyses in living cells of *Arabidopsis* leaves. We used time-lapse microscopy to observe the segmentation of chloroplast budding structures and their transport to the vacuolar lumen. Use of multiple organelle markers revealed that the development of the chloroplast-associated isolation membrane and the formation of RCBs occur simultaneously. These morphological changes did not correspond to the emergence of stromules or to DRP5B-mediated chloroplast division. Therefore, a previously undescribed chloroplast division machinery may be in play during chloroplast autophagy.

## Results

### Chloroplast budding structures containing stroma and envelope are released in response to sugar starvation

To observe the intracellular dynamics of the piecemeal-type of chloroplast autophagy, we subjected the mature leaves of *Arabidopsis* plants to incubation in darkness, as this treatment induces sugar starvation and accelerates leaf senescence, thus activating the RCB-mediated chloroplast autophagy (*Izumi et al., 2010*). Exogenous application of concanamycin A (concA), an inhibitor of vacuolar H⁺-ATPase, allows the vacuolar accumulation of autophagic cargos, including RCBs, facilitating their visualization (*Yoshimoto et al., 2004*; *Ishida et al., 2008*). Following the excision and incubation in darkness of leaves harboring chloroplast stroma–targeted fluorescent protein markers such as RUBISCO SMALL

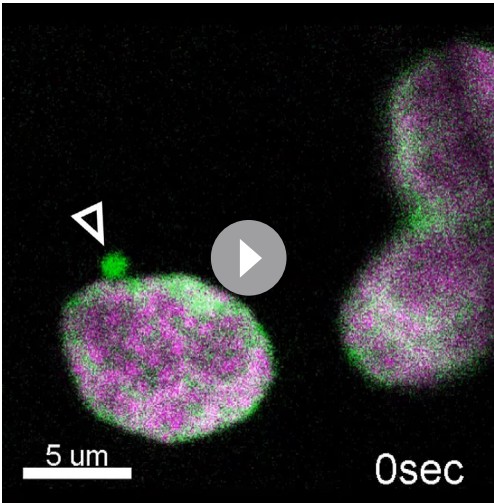

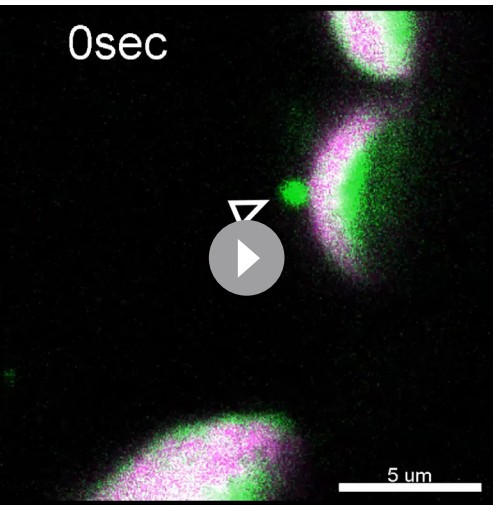

**Video 7.** Tracking of a Rubisco-containing body (RCB) marked by CT-GFP. The second rosette leaf from a 21-day-old plant accumulating chloroplast stroma–targeted GFP (CT-GFP) was incubated in sugar-free solution in darkness for time-lapse monitoring of an RCB marked by CT-GFP. Arrowhead indicates an RCB. Images acquired every 0.52 s are displayed at 20 frames/s. Scale bar, 5 μm. Green, CT-GFP; magenta, chlorophyll fluorescence. Only the video of the merged channels is shown. This video was used to generate *Figure 3A*.

https://elifesciences.org/articles/93232/figures#video7

**Video 8.** Tracking of a Rubisco-containing body (RCB) marked by RBCS-tagRFP. The second rosette leaf from a 22-day-old plant accumulating stromal RBCS-tagRFP was incubated in sugar-free solution in the dark for time-lapse monitoring of an RCB marked by RBCS-tagRFP. Arrowhead indicates an RCB. Images acquired every 1 s are displayed at 20 frames/s. Scale bar, 5 μm. Green, RBCS-tagRFP; magenta, chlorophyll fluorescence. Only the video of the merged channels is shown. This video was used to generate *Figure 3—figure supplement 1A*.

https://elifesciences.org/articles/93232/figures#video8

SUBUNIT (RBCS) fused to monomeric red fluorescent protein (mRFP) in the presence of concA, we observed the accumulation of many small puncta containing the RBCS-mRFP signal in the vacuole, that is, RCBs (*Figure 1—figure supplement 1A, C*). Although co-incubation of dark-treated leaves with concA and mineral nutrient–rich Murashige and Skoog (MS) salts did not block the RCB accumulation, the addition of sucrose did, by rescuing the sugar-starved leaf (*Figure 1—figure supplement 1A, C*). We detected no RCBs when we subjected the leaves of a mutant for the core *ATG* gene *ATG7* to dark incubation in sucrose-free solution (*Figure 1—figure supplement 1B, C*). These observations indicate that RCBs are a type of autophagic cargo involved in the degradation of chloroplast fragments in response to sugar starvation. Therefore, in this study, we monitored chloroplast morphology in sugar-starved leaves resulting from their incubation in darkness in a sugar-free solution.

We wished to monitor the changes in the three-dimensional (3D) morphology of chloroplasts during autophagic degradation of chloroplast fragments. We thus used a two-photon excitation microscope equipped with a confocal spinning-disk unit (*Otomo et al., 2015*), which facilitates the monitoring of changes in the 3D cellular structures of living plant cells (*Sasaki et al., 2019*; *Nakamura et al., 2021b*). Using dark-incubated leaves from transgenic plants harboring chloroplast-targeted *GFP* (*CT-GFP*) or *RBCS-mRFP* transgenes, we determined that mesophyll chloroplasts form a type of budding structure containing stromal components (*Figure 1A, B*, arrowheads). We observed these structures budding off from their associated chloroplasts within a few minutes (*Figure 1A, B*, *Videos 1 and 2*). Dual detection of the chloroplast stromal marker RBCS fused to enhanced yellow fluorescent protein (RBCS-EYFP) and of chlorophyll autofluorescence confirmed that the budding structure specifically contained stroma material (EYFP positive) without any chlorophyll signal, a marker of thylakoid membrane (*Figure 1C*, *Video 3*). These time-lapse observations reveal that chloroplasts form budding structures that are released in sugar-starved leaves. The chloroplast buds are cytoplasmic spherical bodies (approximately 0.5–1.5 μm in diameter) that are labeled by a chloroplast stroma marker and are adjacent to a main chloroplast body.

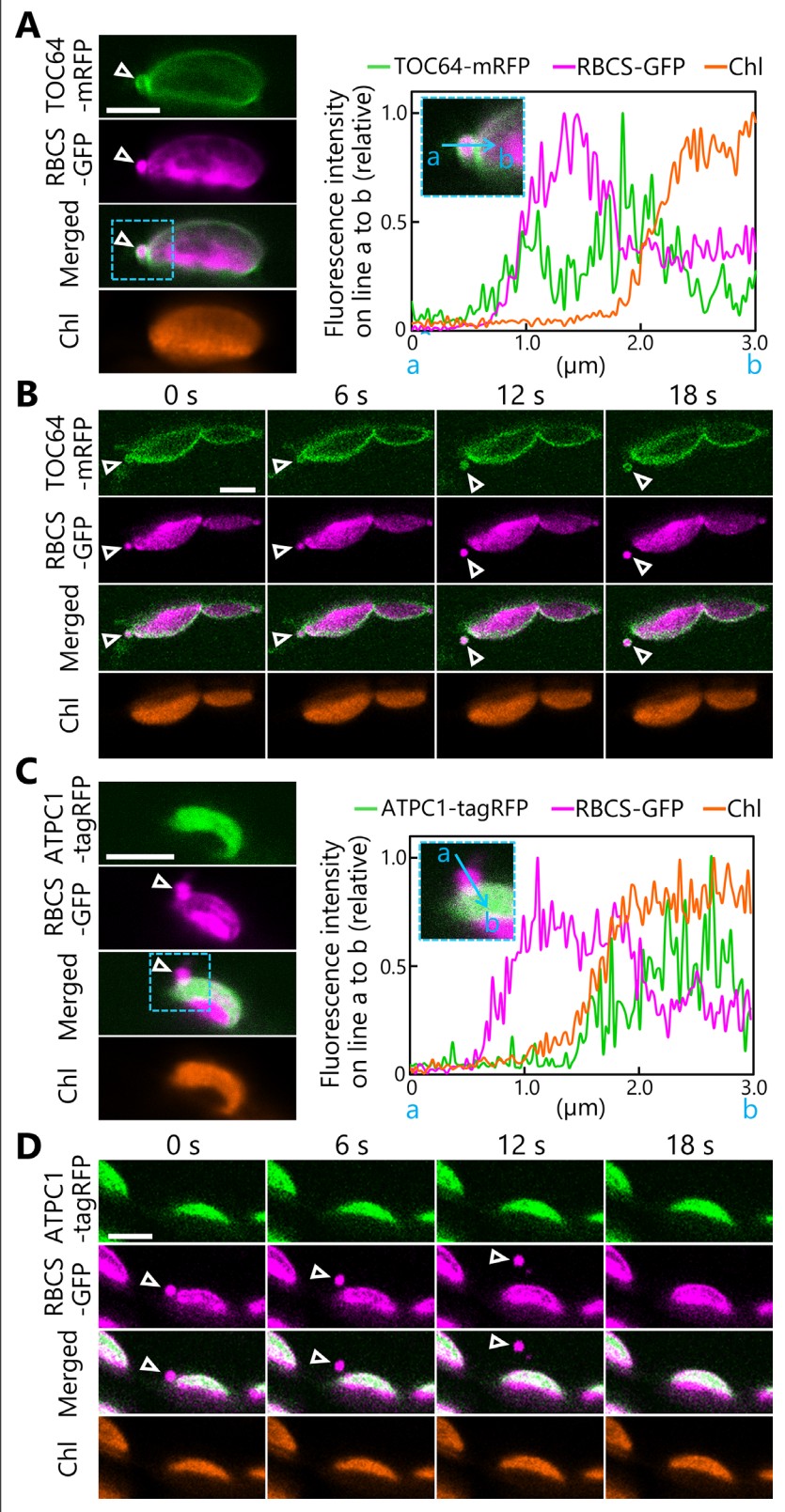

**Figure 2.** Chloroplast buds containing stroma and envelope components are released from the chloroplasts. Time-lapse observations of *Arabidopsis* mesophyll cells accumulating the chloroplast stroma marker along with an envelope marker or a thylakoid membrane marker. Leaves accumulating stromal RBCS-GFP along with outer envelope–bound TOC64-mRFP (**A, B**) or with thylakoid membrane–bound ATPC1-tagRFP (**C, D**) were incubated in

*Figure 2 continued on next page*

*Figure 2 continued*

sugar-free solution in darkness for 5–9 hr from dawn and then observed. Second rosette leaves from 21- to 22-day-old plants were used. Images in (**B**) or (**D**) are still frames from *Videos 4 and 6*, respectively. Time scales above the images indicate the elapsed time from the start of the respective videos. Arrowheads indicate chloroplast budding structures. Scale bars, 5 µm. Green, TOC64-mRFP or ATPC1-tagRFP; magenta, RBCS-GFP; orange, chlorophyll (Chl) fluorescence. The graphs in (**A, C**) show fluorescence intensities along the blue lines (a to b) in the magnified images of the area indicated by dashed blue boxes. The intensities are shown relative to the maximum intensity for each fluorescence channel, set to 1.

The online version of this article includes the following source data and figure supplement(s) for figure 2:

**Source data 1.** Source data for the graphs in *Figure 2*.

**Figure supplement 1.** Chloroplast buds contain stroma and envelope proteins.

**Figure supplement 1—source data 1.** Source data for the graphs in *Figure 2—figure supplement 1*.

**Figure supplement 2.** A released chloroplast bud contains an inner envelope marker protein.

---

To estimate the impact of the release of chloroplast buds on the decline in chloroplast volume, we measured the volume of RBCS-mRFP signals in wild-type (WT) and *atg7* leaves before and after incubation in the dark for 24 hr (*Figure 1—figure supplement 2*). Chloroplasts accumulate starch during the day and consume most of this starch during the night (*Smith and Stitt, 2007*). To minimize the effects of starch metabolism on the changes in chloroplast volume, we began sugar-starvation treatment at the end of the night when a major portion of the starch has been consumed. After the treatment, stroma volume decreased in both genotypes (*Figure 1—figure supplement 2B*). However, the decline in volume was smaller in *atg7* leaves (14%) than in WT leaves (30%). The vacuolar transport of chloroplast budding structures as RCBs did not occur in *atg7* leaves (*Figure 1—figure supplement 1*). These results suggest that the release of chloroplast buds contributes to the decrease in chloroplast stroma volume in sugar-starved leaves.

We then performed a simultaneous detection of three organelle markers using conventional confocal microscopes to determine if the chloroplast buds contained membrane-bound proteins. Accordingly, we incubated leaves from plants accumulating the chloroplast outer envelope marker TRANSLOCON AT THE OUTER MEMBRANE OF CHLOROPLASTS 64 (TOC64) fused to mRFP and the stromal marker RBCS-GFP in darkness as described above. Under these conditions, we observed TOC64-mRFP signal surrounding a chloroplast bud (*Figure 2A*, arrowheads, *Figure 2—figure supplement 1A*). We then monitored chloroplasts in leaves accumulating the chloroplast inner envelope membrane protein K$^+$ EFFLUX ANTIPORTER 1 (KEA1) fused to mRFP along with RBCS-GFP after dark incubation. The chloroplast buds contained KEA1-mRFP signal but no chlorophyll signal (*Figure 2—figure supplement 1B*). We also captured the transport of an RCB with TOC64-mRFP or KEA1-mRFP signal (*Figure 2B*, *Video 4*, *Figure 2—figure supplement 2*, *Video 5*). These results indicate that the budding structures contained chloroplast envelope components. In another set of experiments, we generated transgenic plants accumulating the thylakoid membrane marker ATP synthase gamma subunit (ATPC1) fused to tagRFP and stromal RBCS-GFP. The ATPC1-tagRFP fluorescent signal fully overlapped with that of chlorophyll autofluorescence, with both originating from the thylakoid membrane (*Figure 2C, D*). In

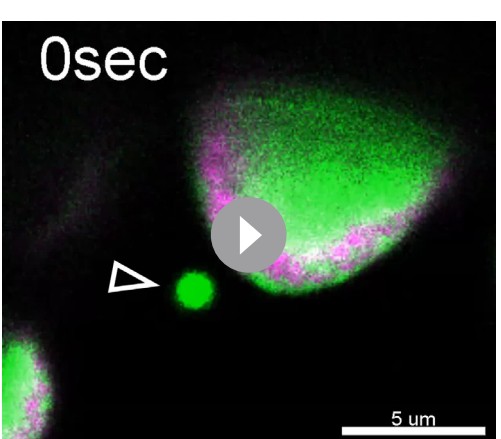

**Video 9.** Tracking of a Rubisco-containing body (RCB) marked by RBCS-EYFP. The second rosette leaf from a 22-day-old plant accumulating stromal RBCS-EYFP was incubated in sugar-free solution in the dark for time-lapse monitoring of an RCB marked by RBCS-EYFP. Arrowhead indicates an RCB. Images acquired every 1 s are displayed at 10 frames/s. Scale bar, 5 µm. Green, RBCS-EYFP; magenta, chlorophyll fluorescence. Only the video of the merged channels is shown. This video was used to generate *Figure 3—figure supplement 1B*.

https://elifesciences.org/articles/93232/figures#video9

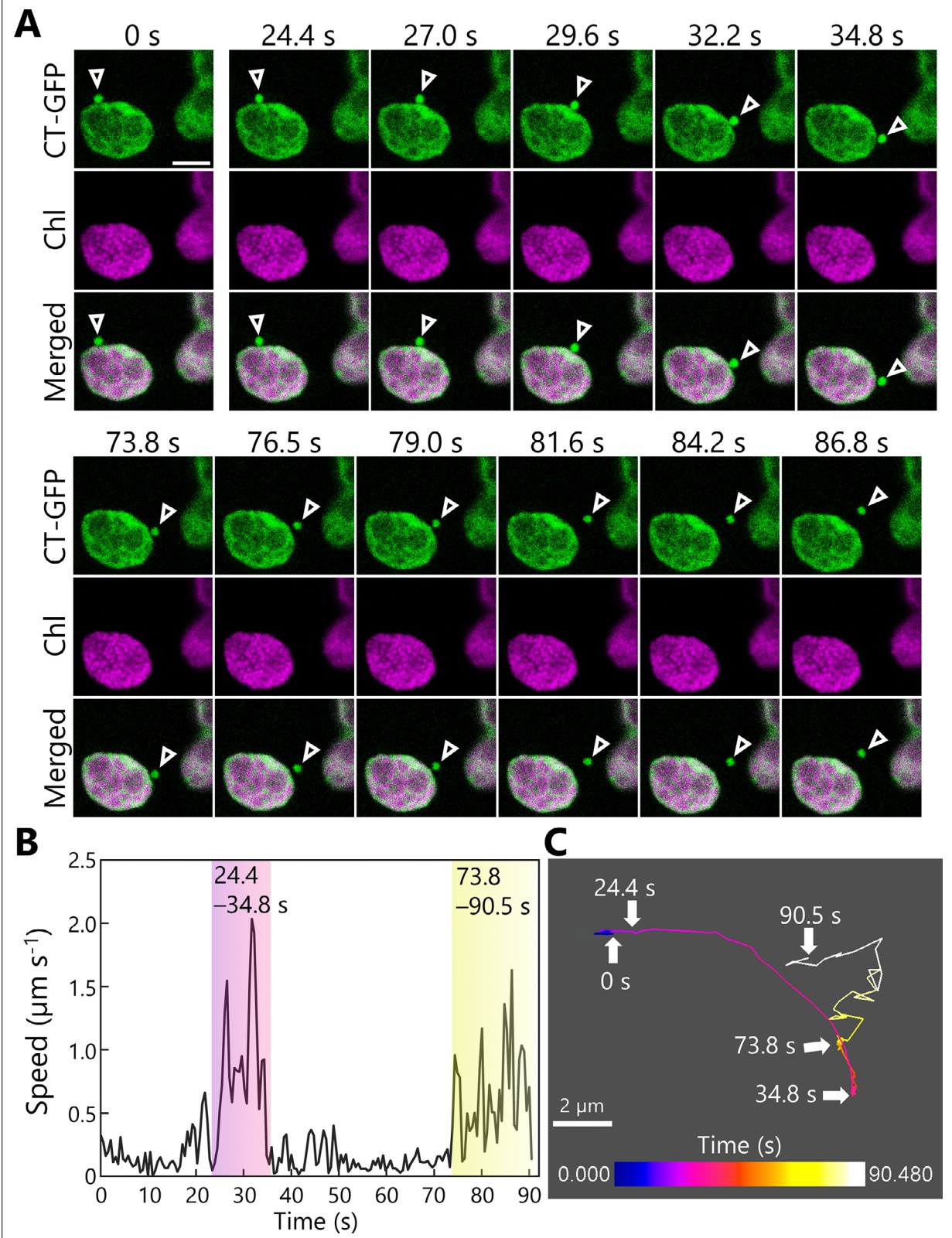

**Figure 3.** Tracking the transport of a Rubisco-containing body (RCB). A leaf accumulating chloroplast stroma–targeted GFP (CT-GFP) was incubated in sugar-free solution in darkness for 6 hr from dawn, and the transport of an RCB was tracked. The second rosette leaf from a 21-day-old plant was used. (**A**) Confocal images during the periods when the RCB moved quickly (24.4–34.8 and 73.8–90.5 s). Arrowheads indicate an RCB. The images are still frames from *Video 7*. Time scales above the images indicate the elapsed time from the start of the video. Green, CT-GFP; magenta, chlorophyll (Chl)

*Figure 3 continued on next page*

*Figure 3 continued*

fluorescence. Scale bar, 5 µm. (**B**) Calculated speed of the tracked RCB in (**A**). (**C**) The track of the RCB. The color of the track line changes over time, as indicated by the color bar. Scale bar, 2 µm.

The online version of this article includes the following source data and figure supplement(s) for figure 3:

**Source data 1.** Source data for the graph in *Figure 3*.

**Figure supplement 1.** Rubisco-containing bodies (RCBs) appear to begin random movement during their tracking.

the chloroplast budding structures, however, we detected no ATPC1-tagRFP signal, in contrast to the strong stromal RBCS-GFP signal (*Figure 2C*, arrowheads, *Figure 2—figure supplement 1C*). In agreement with this observation, we captured a released RCB that did not contain any ATPC1-tagRFP or chlorophyll signals, but accumulated plentiful RBCS-GFP (*Figure 2D*, *Video 6*). These observations support the notion that the chloroplast buds containing stromal and envelope proteins are released as RCBs without thylakoid membranes. These live-cell imaging results are consistent with an immuno-electron microscopy study of RCBs in wheat (*Triticum aestivum*) leaves (*Chiba et al., 2003*).

## The vacuolar membrane dynamically interacts with chloroplast stromal components to incorporate them into the vacuolar lumen

We tracked the trafficking of RCBs in mesophyll cells further and obtained a two-dimensional sequential series of images of the release and transport of an RCB in leaves accumulating stromal CT-GFP (*Figure 3A*, *Video 7*). This time series included two phases during which the punctum moved quickly (*Figure 3B*, 24.4–34.8 and 73.8–90.5 s). A tracking analysis for the RCB indicated that during the early phase (24.4–34.8 s), the punctum moved rather smoothly in one general direction (*Figure 3C*, *Video 7*); however, during the later phase (73.8–90.5 s), the punctum appeared to move more randomly, akin to Brownian movement (*Figure 3C*, *Video 7*). Other tracking series of dark-incubated leaves accumulating RBCS-tagRFP or RBCS-EYFP also showed RCBs starting to exhibit random movement: after 185 s in *Figure 3—figure supplement 1A*; *Video 8* or after 25 s in *Figure 3—figure supplement 1B*; *Video 9*. Previous studies showed that autophagic bodies accumulating into the vacuolar lumen exhibit such random movements (*Ishida et al., 2008*). We thus reasoned that the incorporation of RCBs into the vacuolar lumen from the cytoplasm takes place when these structures start to exhibit random movement.

We therefore turned our attention to the dynamics of the vacuolar membrane when it is incorporating RCBs into the vacuolar lumen in transgenic plants accumulating the vacuolar membrane marker VACUOLAR H$^+$-PYROPHOSPHATASE 1 (VHP1) fused to mGFP (*Segami et al., 2014*) along with the stromal marker RBCS-mRFP (*Figure 4*). From one of our time-lapse imaging series, we identified still frames of an RCB present in the cytoplasm being engulfed by the vacuolar membrane (*Figure 4A*, open arrowheads from 25.4 to 30.5 s, *Video 10*). Later in the time series, we observed the opening of the part of the vacuolar membrane that had engulfed the RCB, resulting in the RCB being released into the vacuolar lumen (*Figure 4A*, filled arrowheads from 47.0 to 52.0 s, *Video 10*). We captured another instance of the dynamics of the vacuolar membrane in another time series, with the vacuolar membrane first surrounding the RCB (52.6–57.4 s) before releasing it (65.3–72.9 s) into the vacuolar lumen (*Figure 4B*, *Video 11*). *Figure 4—figure supplement 1* and *Videos 12–14* show similar membrane dynamics that were observed in three individual plants. An RCB marked by RBCS-mRFP was engulfed by the vacuolar membrane at 48.0–54.0 s (*Figure 4—figure supplement 1A*, *Video 12*), 6.0–15.0 s (*Figure 4—figure supplement 1B*, *Video 13*), and 111.0–123.0 s (*Figure 4—figure supplement 1C*, *Video 14*). The RCBs were then released into the vacuolar lumen. Our time-lapse imaging approach thus allowed us to successfully visualize the cytoplasm-to-vacuole transport progression of chloroplast stroma material via autophagy.

## Formation and segmentation of chloroplast buds occur simultaneously with autophagosome maturation

Based on the observation of spherical structures budding off from chloroplasts (*Figures 1–3*), we hypothesized that these structures are formed in response to sugar starvation and are subsequently recognized and sequestered by the autophagosomal membranes. We thus expected the leaves of autophagy-deficient mutants to accumulate multiple extended structures of chloroplast stroma as

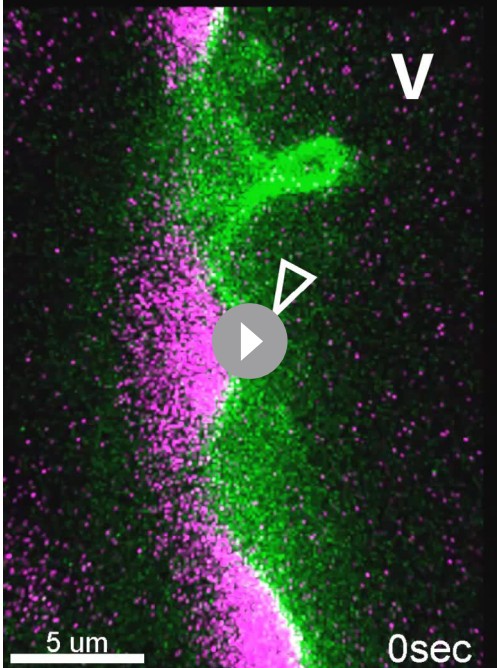

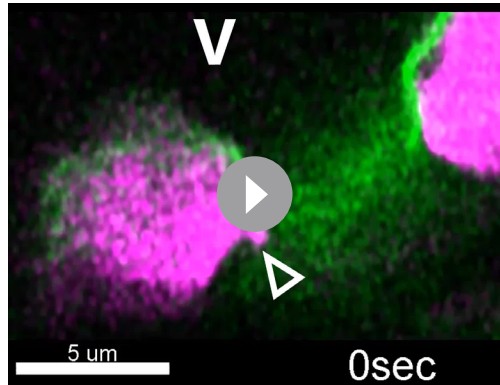

**Video 11.** Video 2 of the vacuolar incorporation of a Rubisco-containing body (RCB). Additional time-lapse data for the experiment described in *Figure 4* and *Video 10*. The second rosette leaf from a 23-day-old plant was used. Arrowhead indicates an RCB incorporated into the vacuolar lumen. Images acquired every 0.317 s are displayed at 25 frames/s. Scale bar, 5 μm. Green, VHP1-mGFP; magenta, RBCS-mRFP. Only the video of the merged channels is shown. This video was used to generate *Figure 4B*.

https://elifesciences.org/articles/93232/figures#video11

**Video 10.** Incorporation of a Rubisco-containing body (RCB) into the vacuolar lumen. The second rosette leaf from a 21-day-old plant accumulating the chloroplast stroma marker RBCS-mRFP along with the vacuolar membrane marker VHP1-mGFP was incubated in sugar-free solution in darkness for time-lapse imaging. Arrowhead indicates an RCB incorporated into the vacuolar lumen. Images acquired every 1.27 s are displayed at 10 frames/s. Scale bar, 5 μm. Green, VHP1-mGFP; magenta, RBCS-mRFP. Only the video of the merged channels is shown. This video was used to generate *Figure 4A*.

https://elifesciences.org/articles/93232/figures#video10

chloroplast protrusions upon exposure to sugar starvation. This hypothesis seemed consistent with the observations of stromule-rich cells in hypocotyls or dark-incubated leaves of autophagy-deficient *atg5* or *atg7* mutants, compared with non-mutants (*Ishida et al., 2008*; *Spitzer et al., 2015*), assuming that the accumulation of chloroplast protrusions precedes stromule formation in *atg* mutants. To test this hypothesis, we subjected the leaves of *atg5* and *atg7* mutants,

each defective in the function of a core ATG protein, to darkness-induced sugar starvation and treatment with concA for 1 day (*Figure 5*). We then evaluated the appearance of RCBs and chloroplast protrusions comprising extended structures of chloroplast stroma labeled by RBCS-mRFP, approximately 0.5–1.5 μm long. We detected no RCB in the leaves of either mutant (*Figure 5A, B*); however, in both the WT and the mutants, we noticed fewer chloroplast protrusions in dark-incubated leaves treated with concA compared with the respective untreated control leaves, contrary to our hypothesis (*Figure 5C, D*). We checked single mutants of *ATG2* and *ATG10* (*atg2* and *atg10*) to investigate other core ATGs and obtained similar results (*Figure 5—figure*

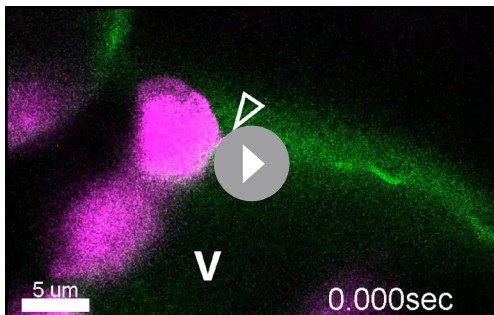

**Video 12.** Video 3 of the vacuolar incorporation of a Rubisco-containing body (RCB). Additional time-lapse data for the experiment described in *Figure 4* and *Video 10*. Arrowhead indicates an RCB incorporated into the vacuolar lumen. Images acquired every 1.5 s are displayed at 10 frames/s. Scale bar, 5 μm. Green, VHP1-mGFP; magenta, RBCS-mRFP. Only the video of the merged channels is shown. This video was used to generate *Figure 4—figure supplement 1A*.

https://elifesciences.org/articles/93232/figures#video12

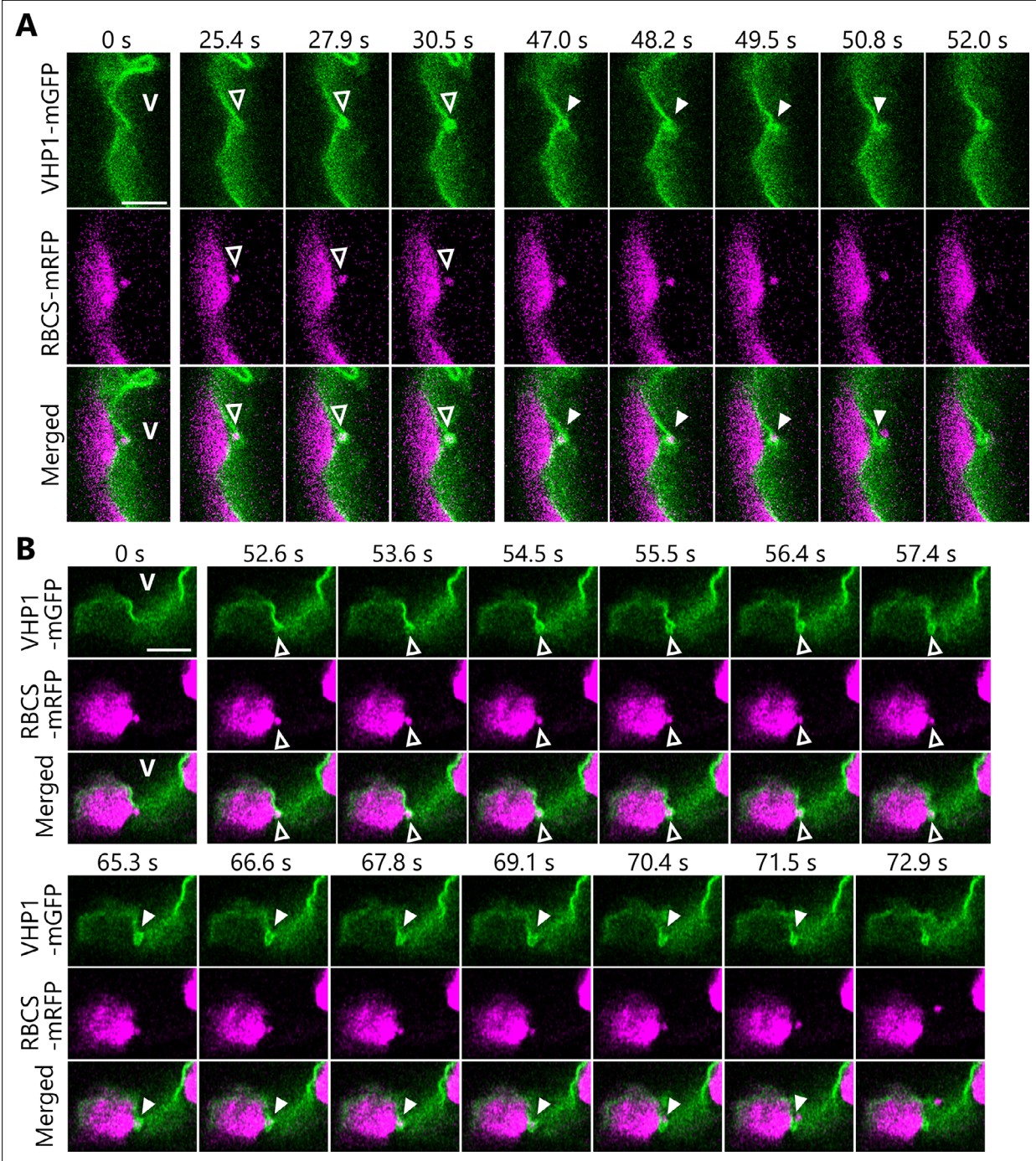

**Figure 4.** Dynamics of the vacuolar membrane during the incorporation of Rubisco-containing bodies (RCBs). Leaves accumulating the chloroplast stroma marker RBCS-mRFP along with the vacuolar membrane marker VHP1-mGFP were incubated in sugar-free solution in darkness for 6–8 hr from dawn, and the behavior of cytosolic RCBs was monitored. Second rosette leaves from 21- or 23-day-old plants were used. The images when the vacuolar membrane engulfs an RCB (25.4–30.5 s in **A** and 52.6–57.4 s in **B**) and when an RCB is incorporated into the vacuolar lumen (47.0–52.0 s in **A** and 65.3–72.9 s in **B**) are shown. The images in (**A**) and (**B**) are still frames from *Videos 10 and 11*, respectively. Time scales above the images indicate the elapsed time from the start of the respective videos. Open arrowheads indicate an RCB engulfed by the vacuolar membrane. Closed arrowheads indicate the open site of the vacuolar membrane for the release of an RCB into vacuolar lumen. V indicates the region of the vacuolar lumen. Green, VHP1-mGFP; magenta, RBCS-mRFP. Scale bars, 5 μm.

The online version of this article includes the following figure supplement(s) for figure 4:

**Figure supplement 1.** Additional observations of the incorporation of Rubisco-containing bodies (RCBs) into the vacuolar lumen.

*supplement 1*): The number of chloroplast protrusions did not increase in *atg2* or *atg10* leaves during a 1-day incubation in darkness.

We therefore posited that autophagosome formation might be required for the production of chloroplast buds. To test this idea, we observed the behavior of autophagosomes in transgenic plants accumulating the isolation membrane marker AUTOPHAGY8a (ATG8a) fused to GFP (as GFP-ATG8a) along with stromal RBCS-mRFP or chloroplast stroma–targeted (CT)-DsRed (*Figure 6*). Following incubation of leaves in darkness and confocal microscopy observation, we observed isolation membranes (marked by GFP-ATG8a) that were tightly associated with the chloroplast surface before any budding structure was visible (*Figure 6A, B*, 0 s). Over time, however, fluorescence monitoring of the same area revealed that the isolation membrane–associated site from the chloroplast side gradually protruded and eventually formed a spherical body containing RBCS-mRFP or CT-DsRed and surrounded by GFP-ATG8a signal, that is, an autophagosome containing an RCB (*Figure 6A, B*, arrowheads; *Videos 15 and 16*).

Using the time-series data shown in *Figure 6B* (*Video 16*), we evaluated the time-dependent changes of the size of the chloroplast bud and the ratio between the major and minor axes of the GFP-ATG8a-labeled autophagosome (*Figure 6C*). This ratio dropped to 1 as the isolation membrane became a spherical autophagosome in the focal plane. During this observation period, the ratio between the major and minor axes started to decrease at around 40 s and approached 1 at around 100 s. The chloroplast budding structure appeared at around 50 s and gradually increased in size until around 120 s. These data support the notion that the chloroplast bud appears as the isolation membrane develops and becomes a segmented RCB when the spherical autophagosome forms. We identified another example in an independent image series (*Figure 6—figure supplement 1*, *Video 17*) where the ratio between the major and minor axes of the isolation membrane started to decrease at approximately 40 s and the chloroplast bud appeared at around 50 s. In the third set of images, we observed the formation of chloroplast buds and later of RCBs near GFP-ATG8a-associated sites on multiple chloroplasts and sequentially (*Figure 6—figure supplement 2*, *Video 18*). Therefore, we conclude that the development of the chloroplast-associated isolation membrane precedes the production of chloroplast buds and of RCBs for piecemeal-type chloroplast autophagy.

We then evaluated the proportion of chloroplast buds associated with GFP-ATG8a signal when leaves accumulating RBCS-mRFP and GFP-ATG8a were incubated in the dark for 5–9 hr from the end of night (*Figure 6—figure supplement 3*). Of the chloroplast buds examined, 64% were GFP-ATG8a positive (*Figure 6—figure supplement 3A–C*). The formation of GFP-ATG8a-labeled chloroplast buds increased in response to the dark treatment (*Figure 6—figure supplement 3D*), but the buds did not appear in leaves from *atg7* plants exposed to dark treatment (*Figure 6—figure supplement 3E*). Therefore, a large portion of chloroplast buds may form synchronously with the development of the isolation membrane under sugar-starvation treatment. We then turned our attention to the isolation membrane and evaluated the proportion of GFP-ATG8a-labeled structures associated with chloroplasts (*Figure 6—figure supplement 4*). Of the isolation membranes examined, 23% localized on the chloroplast surface as flattened structures (*Figure 6—figure supplement 4A, D*), and 20% localized with chloroplast buds as spherical or curved structures (*Figure 6—figure supplement 4B, D*). Therefore, 43% of the GFP-ATG8a-labeled structures participated in chloroplast degradation during the dark treatment in this study. These results support the notion that chloroplast budding accompanied by autophagosome development substantially contributes to the degradation of chloroplast components in sugar-starved leaves.

## SA influences stromule formation in autophagy-deficient mutants

Our initial hypothesis was that stromules are related to RCBs: Stromule-rich cells appeared because of impaired RCB production in *atg* mutants. However, our imaging assays above revealed that RCBs emerge from mesophyll chloroplasts in the absence of prior formation of stromules or protrusions (*Figure 6*), pointing to the existence of another factor responsible for the elevated stromule formation seen in *atg* mutant leaves. A previous study reported the hyperaccumulation of SA in senescing *Arabidopsis atg* mutants (*Yoshimoto et al., 2009*). Since SA signaling stimulates immune responses, including stromule formation (*Caplan et al., 2015*), we speculated that SA might contribute to stromule formation in autophagy-deficient mutants. To investigate this possibility, we generated transgenic plants in the *atg5* mutant background accumulating stromal GFP (CT-GFP) and a construct carrying

NahG, encoding a bacterial SA hydroxylase that catabolizes SA (*Delaney et al., 1994*; *Yoshimoto et al., 2009*). In these plant lines, we evaluated the frequency of chloroplasts forming stromules, which are thin, tubular structures of chloroplast stroma less than 1 µm in diameter (*Hanson and Sattarzadeh, 2011*; *Brunkard et al., 2015*). We examined chloroplasts in guard cells, as this cell type actively forms many more stromules than the mature chloroplasts of mesophyll cells (*Ishida and Yoshimoto, 2008*). Chloroplasts in the guard cells of the *atg5* mutant formed more stromules than WT and NahG-expressing plants (*Figure 7A, B*). The introduction of the NahG construct into the *atg5* mutant background (*atg5* NahG) largely abolished stromule formation (*Figure 7A, B*). To further explore the role of SA in stromule formation, we generated double mutants of *SALICYLIC ACID INDUCTION DEFICIENT 2* (*SID2*), encoding the SA biosynthetic enzyme ISOCHORISMATE SYNTHASE 1, and *ATG5* or *ATG7* (*sid2 atg5* and *sid2 atg7*). The *sid2* mutation also resulted in fewer stromules in the double mutant relative to the respective *atg5* and *atg7* single mutants in guard cells (*Figure 7C, D*).

We next observed stromule formation in mesophyll chloroplasts accumulating CT-GFP or RBCS-mRFP and found that many chloroplasts formed stromules in senescent leaves of 36-day-old *atg5* and *atg7* plants (*Figure 7E, F*, *Figure 7—figure supplement 1*). In cells from the third rosette leaves of 36-day-old plants accumulating CT-GFP, we observed stromules in 22.5% or 15.3% of chloroplasts from the *atg5* or *atg7* single mutants, respectively; however, the introduction of the *sid2* mutation in *sid2 atg5* and *sid2 atg7* led to fewer stromules (9.8% or 2.5% in *sid2 atg5* or *sid2 atg7*, respectively; *Figure 7E, F*). In the leaves of 20-day-old plants, we did not detect stromules in any genotype (*Figure 7—figure supplement 2*). These results support the notion that SA accumulation due to autophagy deficiency activates stromule formation during leaf senescence.

The frequency of stromules was higher in the *sid2 atg5* and *sid2 atg7* double mutants than in WT plants and the *sid2* single mutant (*Figure 7*). We measured the content of hydrogen peroxide ($H_2O_2$) in all plants, since reactive oxygen species (ROS) also trigger stromule formation (*Caplan et al., 2015*). In 36-day-old plants, *atg5* or *atg7* leaves contained 2.7 or 2.8 times more $H_2O_2$ than WT leaves (*Figure 7G*). Notably, the presence of the *sid2* mutation in the *sid2 atg5* and *sid2 atg7* double mutants did not alleviate $H_2O_2$ accumulation (*Figure 7G*). Such ROS accumulation may thus be one factor responsible for the greater number of stromules in *sid2 atg5* and *sid2 atg7* compared with WT and *sid2* plants. Moreover, NahG expression did not counteract the increase in $H_2O_2$ level caused by the *atg5* mutation (*Figure 7—figure supplement 3*). ROS accumulation in *atg* leaves may not depend on stimulation of SA signaling. These results are consistent with the results of histochemical ROS staining of *Arabidopsis* leaves in a previous study (*Yoshimoto et al., 2009*).

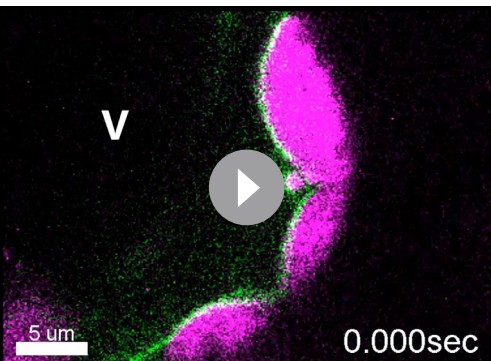

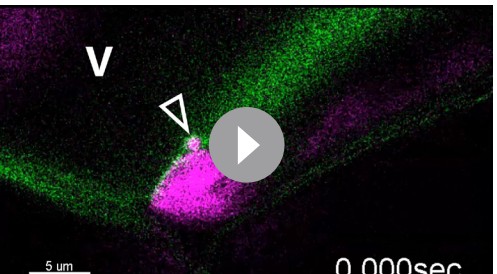

**Video 13.** Video 4 of the vacuolar incorporation of a Rubisco-containing body (RCB). Additional time-lapse data for the experiment described in *Figure 4* and *Video 10*. Arrowhead indicates an RCB incorporated into the vacuolar lumen. Images acquired every 1.5 s are displayed at 10 frames/s. Scale bar, 5 µm. Green, VHP1-mGFP; magenta, RBCS-mRFP. Only the video of the merged channels is shown. This video was used to generate *Figure 4—figure supplement 1B*.

https://elifesciences.org/articles/93232/figures#video13

**Video 14.** Video 5 of the vacuolar incorporation of a Rubisco-containing body (RCB). Additional time-lapse data for the experiment described in *Figure 4* and *Video 10*. Arrowhead indicates an RCB incorporated into the vacuolar lumen. Images acquired every 1.5 s are displayed at 10 frames/s. Scale bar, 5 µm. Green, VHP1-mGFP; magenta, RBCS-mRFP. Only the video of the merged channels is shown. This video was used to generate *Figure 4—figure supplement 1C*.

https://elifesciences.org/articles/93232/figures#video14

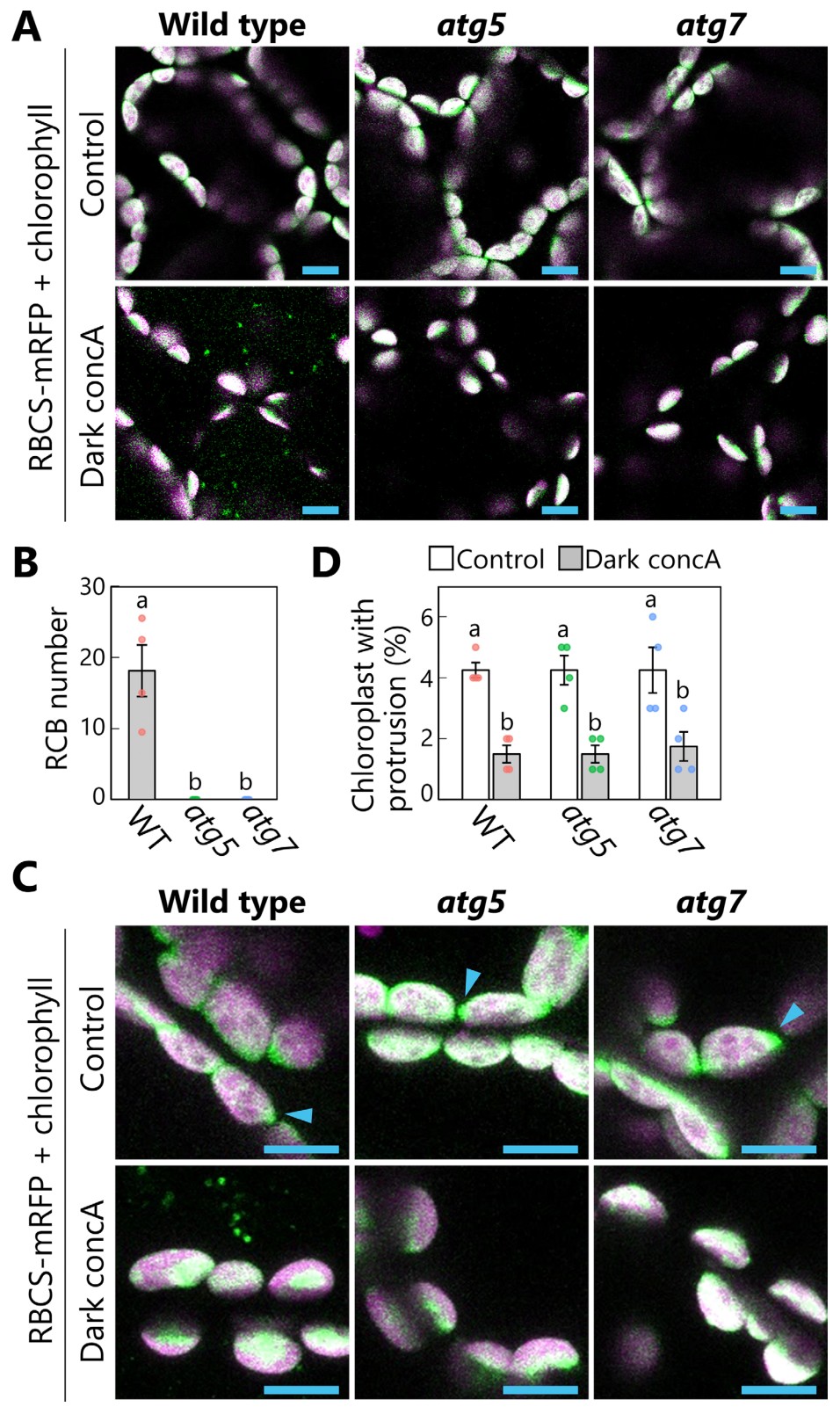

**Figure 5.** Autophagy deficiency does not increase the number of chloroplast protrusions during a 1-day dark treatment. Leaves from wild-type (WT), *atg5*, or *atg7* plants accumulating the chloroplast stroma marker RBCS-mRFP were incubated in sugar-free solution containing 1 μM concanamycin A (concA) for 1 day in darkness. Second rosette leaves from 21-day-old plants were used. Two-dimensional (2D) images of mesophyll cells were

*Figure 5 continued on next page*

*Figure 5 continued*

acquired (**A**), and the number of accumulated Rubisco-containing bodies (RCBs) in the vacuoles was scored (**B**). Small puncta containing RBCS-mRFP without the chlorophyll signal appear green and are RCBs in the vacuole. Leaves from untreated plants are shown as control. The appearance of chloroplast protrusions was observed from orthogonal projections created from z-stack images (15 μm in depth; **C**), and the proportion of chloroplasts forming protrusion structures out of 50 chloroplasts was scored (**D**). Scale bars, 10 μm. Green, RBCS-mRFP; magenta, chlorophyll fluorescence. Only the merged channels are shown. The overlapping regions of RBCS-mRFP and chlorophyll signals appear white. Arrowheads indicate the structures that were counted as a chloroplast protrusion in (**D**). Different lowercase letters denote significant differences based on Tukey's test (p < 0.05). Values are means ± SE (*n* = 4). Dots represent individual data points in each graph.

The online version of this article includes the following source data and figure supplement(s) for figure 5:

**Source data 1.** Source data for the graphs in *Figure 5*.

**Figure supplement 1.** Chloroplast protrusions do not increase in *atg2* or *atg10* mutant leaves during a 1-day dark treatment.

**Figure supplement 1—source data 1.** Source data for the graphs in *Figure 5—figure supplement 1*.

## DRP5B is not required for autophagy of chloroplast fragments

Simultaneous progression of autophagosome formation and organelle segmentation has been previously observed during mitophagy in mammals and budding yeast (*Yamashita et al., 2016*; *Fukuda et al., 2023*). This autophagy-related mitochondrial division does not require the mammalian Drp1- or yeast Dnm1-dependent division machinery, respectively. Here, we examined whether the DRP5B-mediated chloroplast division participates in chloroplast autophagy and RCB production.

To this end, we generated *drp5b* mutant lines accumulating the stromal marker RBCS-mRFP or stromal GFP (CT-GFP). Although the mature leaves of *drp5b* had larger chloroplasts than WT leaves, reflective of the impaired chloroplast division during cell expansion typical of this mutant, many RCBs accumulated when the leaves were incubated in darkness in the presence of concA (*Figure 8A, B*, *Figure 8—figure supplement 1*). We detected no RCB in the leaves of the *drp5b atg5* double mutant (*Figure 8—figure supplement 1*). These results indicate that RCB-mediated chloroplast autophagy is active in the *drp5b* mutant. To obtain an independent confirmation of this result, we performed an immunoblot assay of autophagy flux, based on the detection of free mRFP derived from the vacuolar cleavage of RBCS-mRFP (*Ono et al., 2013*). When we incubated the leaves of plants accumulating RBCS-mRFP in darkness without concA, free mRFP levels increased in leaves of WT but not *atg5* or *atg7* plants (*Figure 8C, D*), indicating the occurrence of autophagy-dependent degradation of stromal proteins in response to sugar starvation in WT plants. The accumulation of free mRFP in the *drp5b* mutant was at least as high as that seen in WT plants (*Figure 8C*). The ratio of free mRFP to RBCS-mRFP increased in WT and *drp5b* plants after dark treatment (6.1 or 4.1 times higher compared with control conditions in WT plants or *drp5b*, respectively), consistent with autophagy-mediated degradation of chloroplast fragments (*Figure 8D*). Confocal microscopy observations of mesophyll cells also showed the spread of mRFP signal in the vacuolar lumen in dark-treated leaves of WT and *drp5b* plants (*Figure 8—figure supplement 2*). These results indicate that the autophagic degradation of chloroplast stroma is active in both genotypes. The accumulation of free mRFP was 3.2 times higher in untreated leaves of *drp5b* compared with untreated WT leaves (*Figure 8D*); therefore, the activity of chloroplast autophagy might be constitutively higher in leaves of *drp5b* than in those of WT plants.

We observed the budding off of chloroplast fragments in leaves of the *drp5b* mutant. We incubated a leaf from the *drp5b* mutant accumulating RBCS-mRFP in darkness, which allowed us to observe the formation of a chloroplast bud and its release (*Figure 9A*, arrowheads; *Video 19*). In a leaf of the *drp5b* mutant accumulating RBCS-mRFP and GFP-ATG8a, we detected the vesiculation of a chloroplast fragment starting at the site to which the isolation membranes, labeled by GFP-ATG8a, were tightly associated (*Figure 9B*). The isolation membranes were anchored at two sites of a large chloroplast in a leaf of the *drp5b* mutant (*Figure 9B*, white and blue arrowheads); the chloroplast gradually protruded before releasing RCBs surrounded by autophagosomes (*Figure 9B*, *Video 20*). We conclude that DRP5B is dispensable for the autophagy-related division of chloroplast fragments.

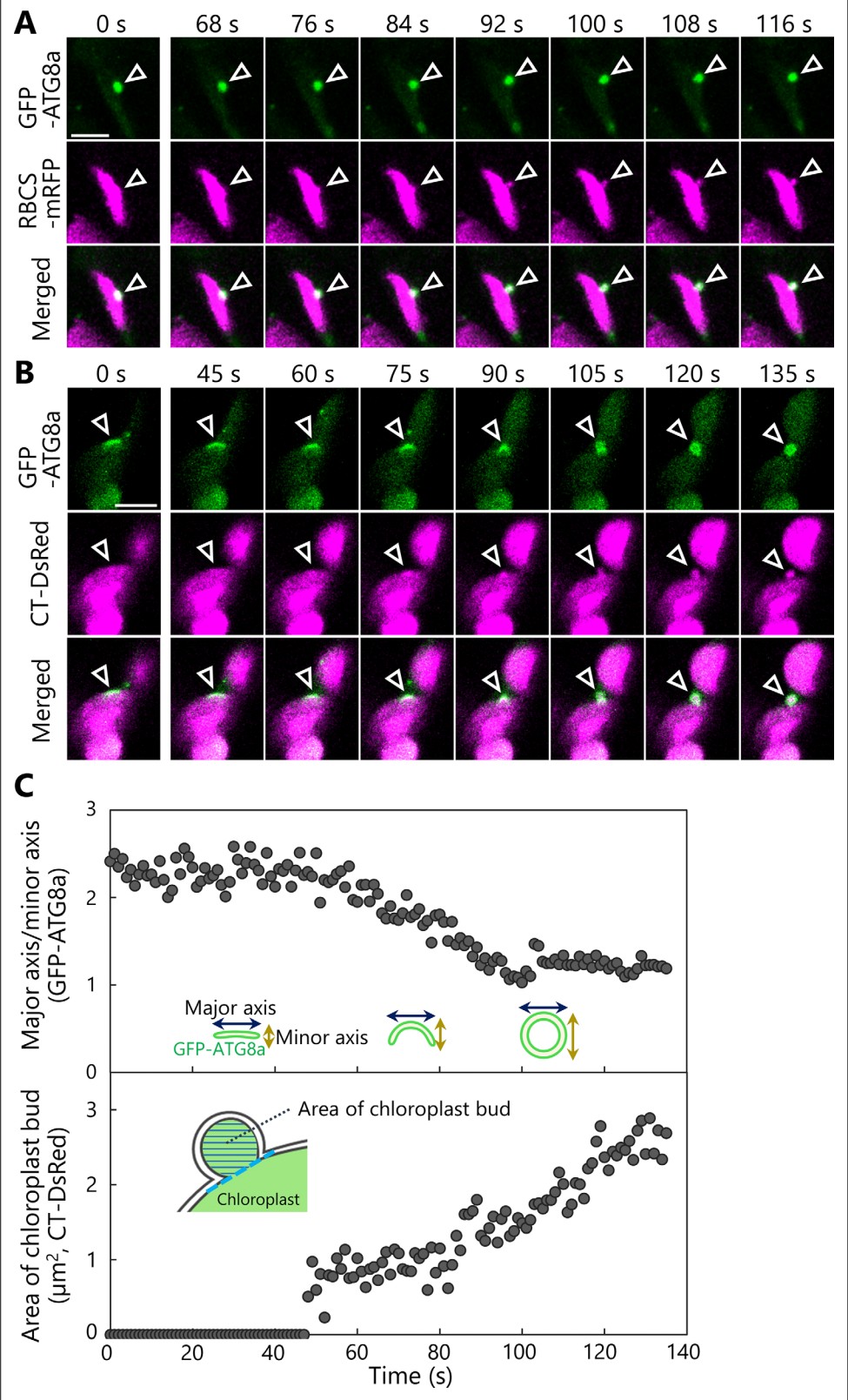

**Figure 6.** The formation of a chloroplast bud and the maturation of the chloroplast-associated isolation membrane occur concomitantly. Leaves accumulating the chloroplast stroma marker, RBCS-mRFP (**A**) or CT-DsRed (**B**), and the isolation membrane marker GFP-ATG8a were incubated in sugar-free solution in darkness for 6 hr from dawn (**A**) or 21 hr from the period in light (**B**) and then observed. Second rosette leaves from 23-day-old plants were used.

*Figure 6 continued on next page*

*Figure 6 continued*

Arrowheads indicate the position of the chloroplast-associated isolation membrane. Images in (**A**) and (**B**) are still frames from *Videos 15 and 16*, respectively. Time scales above the images indicate the elapsed time from the start of the respective videos. Green, GFP-ATG8a; magenta, RBCS-mRFP or CT-DsRed. Scale bars, 5 μm. (**C**) Time-dependent changes in the ratio of the major axis to the minor axis in the GFP-ATG8a-labeled isolation membrane (top) or in the area of the chloroplast budding structure (bottom) as measured from the images in (**B**).

The online version of this article includes the following source data and figure supplement(s) for figure 6:

**Source data 1.** Source data for the graphs in *Figure 6*.

**Figure supplement 1.** Another observation of the budding of the isolation membrane–associated site in a chloroplast.

**Figure supplement 1—source data 1.** Source data for the graphs in *Figure 6—figure supplement 1*.

**Figure supplement 2.** Chloroplast buds surrounded by the isolation membrane appear in multiple chloroplasts.

**Figure supplement 3.** Proportion of chloroplast buds engulfed by GFP-ATG8a-labeled isolation membranes.

**Figure supplement 3—source data 1.** Source data for the graphs in *Figure 6—figure supplement 3*.

**Figure supplement 4.** Proportion of GFP-ATG8a-labeled structures that are associated with chloroplasts.

**Figure supplement 4—source data 1.** Source data for the graph in *Figure 6—figure supplement 4*.

## Discussion

### Chloroplast division machinery functioning in piecemeal-type chloroplast autophagy

We previously demonstrated that chloroplast components in mesophyll cells are transported to the vacuole as a type of autophagic cargo termed RCBs to facilitate amino acid recycling (*Ishida et al., 2008*; *Hirota et al., 2018*). However, how a portion of chloroplasts is mobilized into the vacuolar lumen remained uncertain. In this study,

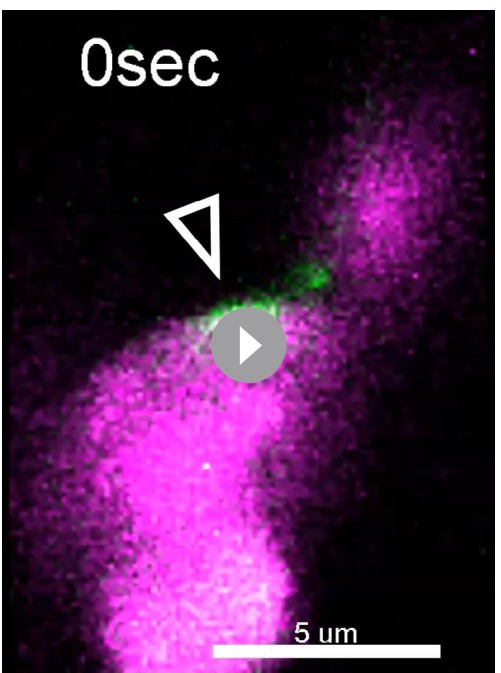

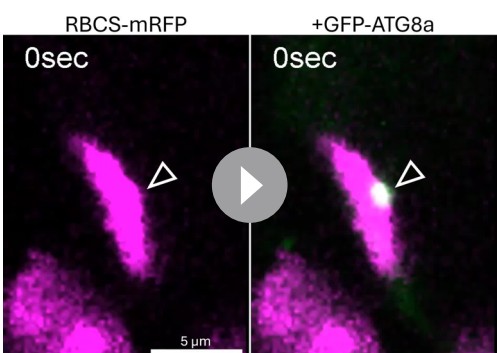

**Video 15.** Budding of the isolation membrane–associated site within a chloroplast. The second rosette leaf from a 23-day-old plant accumulating the chloroplast stroma marker RBCS-mRFP and the isolation membrane marker GFP-ATG8a was incubated in sugar-free solution in darkness for time-lapse imaging. Arrowheads indicate the portion of the chloroplast-associated isolation membrane. Images acquired every 4 s are displayed at 10 frames/s. Scale bar, 5 μm. Green, GFP-ATG8a; magenta, RBCS-mRFP. The video contains RBCS-mRFP and the merged channels. This video was used to generate *Figure 6A*.
https://elifesciences.org/articles/93232/figures#video15

**Video 16.** Autophagosome development and chloroplast segmentation occur concomitantly. The second rosette leaf from a 23-day-old plant accumulating the chloroplast stroma–targeted DsRed (CT-DsRed) and the isolation membrane marker GFP-ATG8a was incubated in sugar-free solution in darkness for time-lapse imaging. Arrowhead indicates the portion of the chloroplast-associated isolation membrane. Images acquired every 1 s are displayed at 20 frames/s. Scale bar, 5 μm. Green, GFP-ATG8a; magenta, CT-DsRed. Only the video of the merged channels is shown. This video was used to generate *Figure 6B*.
https://elifesciences.org/articles/93232/figures#video16

high-resolution time-lapse imaging techniques allowed us to visualize the trafficking progression of chloroplast fragments from their parental chloroplasts to the vacuolar lumen. Importantly, we established that the development of the chloroplast-associated isolation membrane and the division of the chloroplast fragment occur simultaneously. This autophagy-related chloroplast division does not rely on DRP5B-mediated chloroplast division. Likewise, the formation of chloroplast buds is unlikely to be linked to the formation of stromules. Therefore, an unknown division machinery may be required for the autophagy-related division of chloroplast fragments. A similar organelle division mechanism takes place during mitophagy in mammals and budding yeast. The site of a mitochondrion that is associated with the isolation membrane protrudes and divides as the isolation membrane develops and becomes an enclosed autophagosome (*Yamashita et al., 2016*). This type of mitochondrial division is independent of Drp1 and Dnm1. In yeast, the mitochondrial intermembrane-space protein mitofissin/Atg44 divides the mitochondrial fragment in coordination with autophagosome maturation (*Fukuda et al., 2023*). A functionally equivalent protein has not been identified in mammals.

Numerous studies have explored the molecular mechanisms of chloroplast division during the development of juvenile leaves (*Chen et al., 2018*). For instance, in *Arabidopsis* plants, ARC6 and PARALOG OF ARC6 (PARC6) mediate the tethering of the FTSZ ring on the chloroplast inner envelope (*Vitha et al., 2003*). PLASTID DIVISION 1 (PDV1) and PDV2 recruit DRP5B to the division site on the chloroplast outer envelope (*Miyagishima et al., 2006*). The ARC6–PDV2 and PARC6–PDV1 complexes may control the coordination of the stromal FTSZ ring and the cytosolic DRP5B ring (*Wang et al., 2017*; *Chen et al., 2018*). Therefore, the controlled division of plastids requires the cooperation of multiple protein complexes across the inner and outer envelopes. Nevertheless, whether chloroplasts divide during their piecemeal degradation has not been evaluated. It seems likely that autophagy-related division of chloroplast fragments needs to be carefully regulated, as the size of the segment has to fit into the autophagosome. Since the volume occupied by chloroplasts per cell is tightly regulated during cell expansion (*Pyke and Leech, 1994*), the decline of chloroplast volume

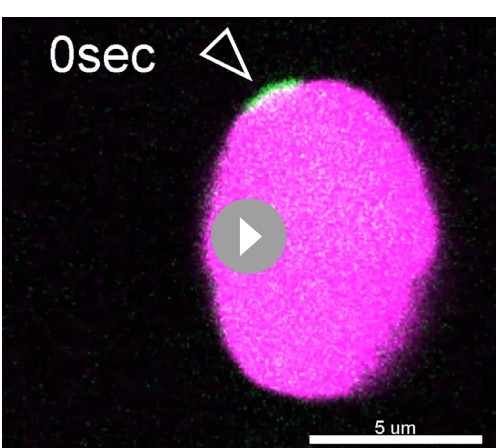

**Video 17.** Another time-lapse assay showing the concomitant progression of autophagosome development and chloroplast segmentation. The second rosette leaf from a 22-day-old plant accumulating the chloroplast stroma–targeted DsRed (CT-DsRed) and the isolation membrane marker GFP-ATG8a was incubated in sugar-free solution in darkness for time-lapse imaging. Arrowhead indicates the portion of the chloroplast-associated isolation membrane. Images acquired every 2 s are displayed at 10 frames/s. Scale bar, 5 µm. Green, GFP-ATG8a; magenta, CT-DsRed. Only the video of the merged channels is shown. This video was used to generate *Figure 6—figure supplement 1*.

https://elifesciences.org/articles/93232/figures#video17

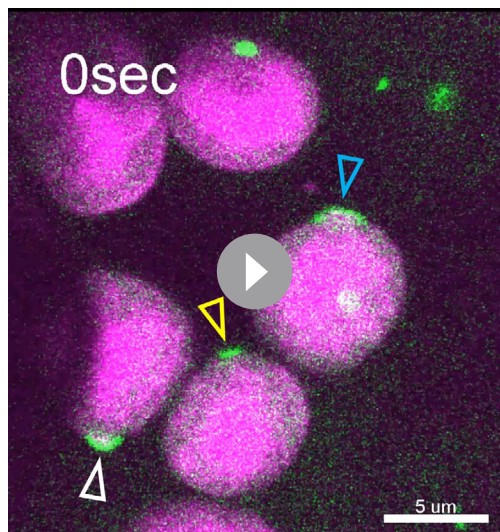

**Video 18.** Autophagy-related chloroplast segmentation occurs in sequence. The second rosette leaf from a 21-day-old plant accumulating the chloroplast stroma marker RBCS-tagRFP and the isolation membrane marker GFP-ATG8a was incubated in sugar-free solution in darkness for time-lapse imaging. White, yellow, and blue arrowheads indicate the isolation membrane–associated sites in different chloroplasts. Images acquired every 2 s are displayed at 10 frames/s. Scale bar, 5 µm. Green, GFP-ATG8a; magenta, RBCS-tagRFP. Only the video of the merged channels is shown. This video was used to generate *Figure 6—figure supplement 2*.

https://elifesciences.org/articles/93232/figures#video18

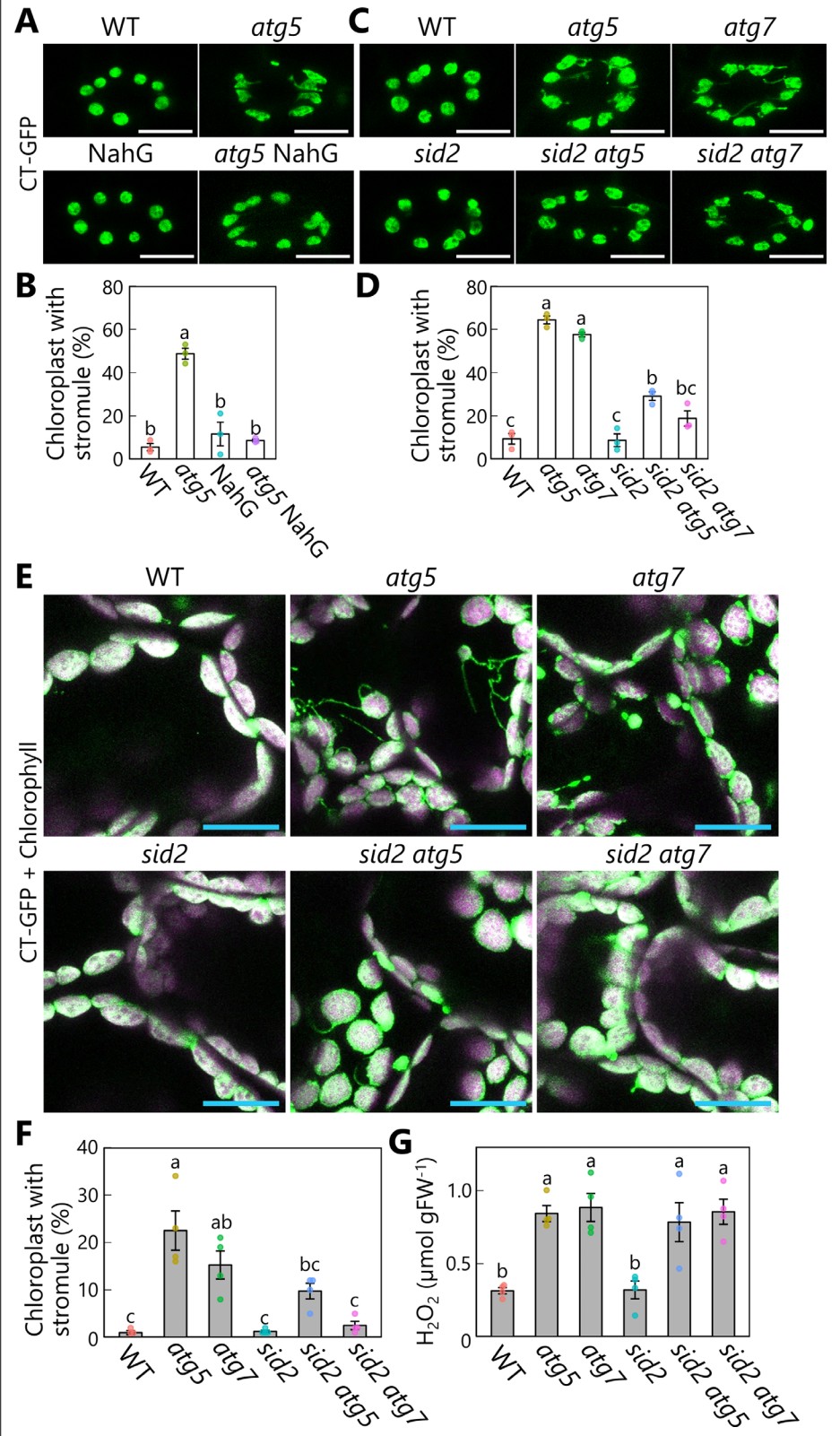

**Figure 7.** Diminished salicylic acid signal suppresses stromule formation in autophagy-deficient mutants.
(**A**) Orthogonal projections produced from z-stack images (10 μm in depth) of guard cells from wild-type (WT),
*atg5*, NahG, and *atg5* NahG leaves accumulating chloroplast stroma−targeted GFP (CT-GFP). Second rosette
leaves from 13-day-old plants were used. Scale bars, 10 μm. (**B**) Percentage of chloroplasts forming stromules in 10

*Figure 7 continued on next page*

*Figure 7 continued*

stomata, from the observations described in (**A**). (**C**) Orthogonal projections produced from *z*-stack images (10 μm in depth) of guard cells from WT, *atg5*, *atg7*, *sid2*, *sid2 atg5*, and *sid2 atg7* leaves accumulating CT-GFP. Scale bars, 10 μm. (**D**) Percentage of chloroplasts forming stromules in 10 stomata, from the observations described in (**C**). (**E**) Orthogonal projections produced from *z*-stack images (15 μm in depth) of mesophyll cells from WT, *atg5*, *atg7*, *sid2*, *sid2 atg5*, and *sid2 atg7* leaves accumulating CT-GFP. Third rosette leaves from 36-day-old plants were observed. Green, CT-GFP; magenta, chlorophyll fluorescence. Only the merged channels are shown. Scale bars, 20 μm. (**F**) Percentage of chloroplasts forming stromules out of 50 chloroplasts in mesophyll cells, from the observations described in (**E**). (**G**) Hydrogen peroxide ($H_2O_2$) content in third rosette leaves from 36-day-old WT, *atg5*, *atg7*, *sid2*, *sid2 atg5*, and *sid2 atg7* plants. Different lowercase letters denote significant differences based on Tukey's test ($p < 0.05$). Values are means ± SE ($n = 3$ in **B** and **D** or 4 in **F** and **G**). Dots represent individual data points in each graph.

The online version of this article includes the following source data and figure supplement(s) for figure 7:

**Source data 1.** Source data for the graphs in *Figure 7*.

**Figure supplement 1.** Autophagy deficiency activates stromule formation from mesophyll chloroplasts accumulating RBCS-mRFP in senescing leaves.

**Figure supplement 1—source data 1.** Source data for the graph in *Figure 7—figure supplement 1*.

**Figure supplement 2.** Autophagy deficiency does not activate stromule formation from mesophyll chloroplasts in young leaves.

**Figure supplement 3.** NahG expression does not counteract the increase in hydrogen peroxide ($H_2O_2$) content produced by the *atg5* mutation.

**Figure supplement 3—source data 1.** Source data for the graph in *Figure 7—figure supplement 3*.

---

during starvation or senescence might also be highly controlled. Further studies are needed to elucidate the molecular machinery underlying the autophagy-related division of chloroplast fragments for degradation.

In our evaluation of chloroplast buds associated with GFP-ATG8a in excised leaves exposed to darkness (*Figure 6—figure supplement 3*), 36% of chloroplast budding structures were not labeled by GFP-ATG8a. Since endogenous ATG8 family members (ATG8a–8i) were functional, such structures might have formed due to the activities of non-labeled ATG8 proteins. Another possibility is that autophagy-independent chloroplast budding might occur. For example, stromule-related structures might form. CHLOROPLAST VESICULATION (CV)–containing vesicles are also observed as autophagy-independent puncta released from chloroplasts for degradation (*Wang and Blumwald, 2014*; *Pan et al., 2023*). However, biochemical assays for chloroplast autophagy flux suggested that ATG-dependent pathways were the main routes for the vacuolar degradation of stromal components during the darkness-induced sugar-starvation treatment used in this study (*Figure 8*). Previous studies have revealed that CV plays a role in plant responses to abiotic stress, such as drought or salt stress (*Wang and Blumwald, 2014*; *Pan et al., 2023*). A recent study suggested that autophagy, but not CV-dependent pathways, is the main contributor to the adaptative response of metabolism to sugar starvation (*Barros et al., 2023*). Overall, these insights support the notion that, during the early phase of sugar starvation (such as 1–2 days of extended darkness), the autophagy-associated machinery is mainly responsible for chloroplast segmentation and degradation. It is unclear what type of process contributed to the decrease in chloroplast stroma volume in leaves of *atg7* plants exposed to dark treatment (*Figure 1—figure supplement 2*).

An assay of autophagic flux of RBCS-mRFP protein in leaves from *drp5b* mutants suggested the possibility that chloroplast autophagy activity is constitutively higher in leaves of *drp5b* than in WT plants (*Figure 8*). The impaired chloroplast division due to the *drp5b* mutation might affect photosynthesis-mediated energy production and reduce energy availability, thereby activating chloroplast autophagy without dark treatment. As described in the Introduction, entire organelle–type autophagy (chlorophagy) mediates the degradation of chloroplasts in their entirety. Mesophyll cells in leaves of *drp5b* mutants contain few chloroplasts (approximately 4 per cell), whereas those in WT leaves contain approximately 60 chloroplasts (*Miyagishima et al., 2006*). Since the volume in a cell occupied by chloroplasts is similar between genotypes, chloroplasts from *drp5b* are much larger than WT chloroplasts. Chloroplasts that are too large might show inhibited chlorophagy, resulting in the alternative activation of piecemeal-type chloroplast autophagy. Analyzing how chloroplasts are

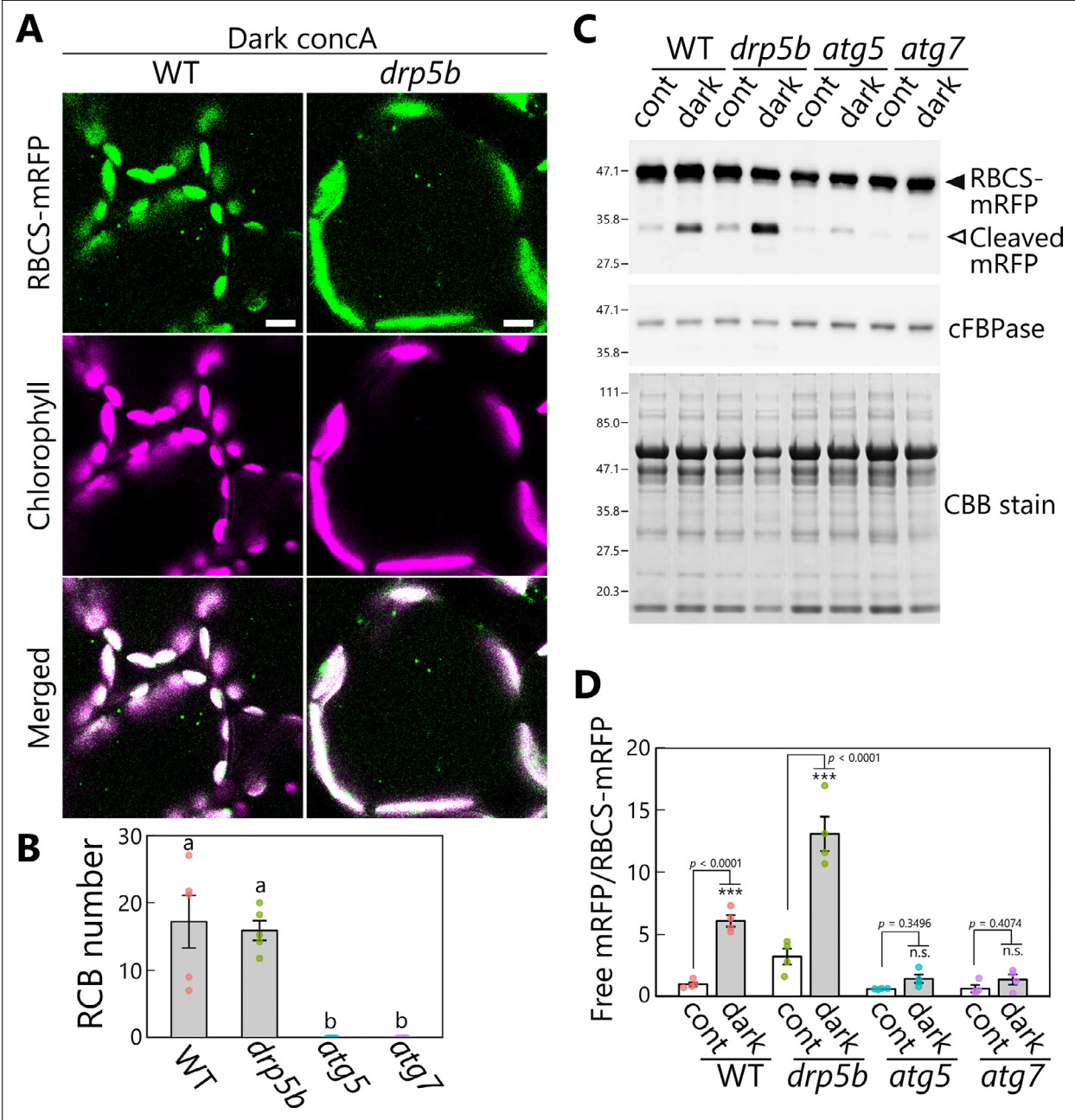

**Figure 8.** DRP5B is dispensable for chloroplast autophagy in sugar-starved leaves. (**A**) Confocal images of mesophyll cells from wild-type (WT) and *drp5b* leaves accumulating the stroma marker RBCS-mRFP. Second rosette leaves from 21-day-old plants were incubated in sugar-free solution containing 1 μM concanamycin A (concA) in darkness for 1 day. Green, RBCS-mRFP; magenta, chlorophyll fluorescence. Scale bars, 10 μm. (**B**) Number of accumulated Rubisco-containing bodies (RCBs) in WT, *drp5b*, *atg5*, and *atg7* leaves, counted from the observations of leaves incubated in the dark with concA described in (**A**). Different lowercase letters denote significant differences based on Tukey's test (p < 0.05). Values are means ± SE (*n* = 4–5). (**C**) Biochemical detection of autophagy flux for chloroplast stroma based on a free RFP assay. RFP and cFBPase (loading control) were detected by immunoblotting of soluble protein extracts from leaves of WT, *drp5b*, *atg5*, and *atg7* plants accumulating RBCS-mRFP. Protein extracts from either untreated control leaves (cont) or leaves after 2 days of incubation in darkness (dark) were used. Total protein was detected by Coomassie Brilliant Blue (CBB) staining as a loading control. The filled arrowhead indicates RBCS-mRFP fusion, and the open arrowhead indicates free mRFP derived from the cleavage of RBCS-mRFP. (**D**) Quantification of the free mRFP/RBCS-mRFP ratio shown relative to that of untreated WT plants, which was set to 1. Asterisks denote significant differences based on *t*-test (***p < 0.001; n.s., not significant). Values are means ± SE (*n* = 4). Dots represent individual data points in each graph.

The online version of this article includes the following source data and figure supplement(s) for figure 8:

**Source data 1.** Original files for western blot analysis displayed in *Figure 8C*.

*Figure 8 continued on next page*

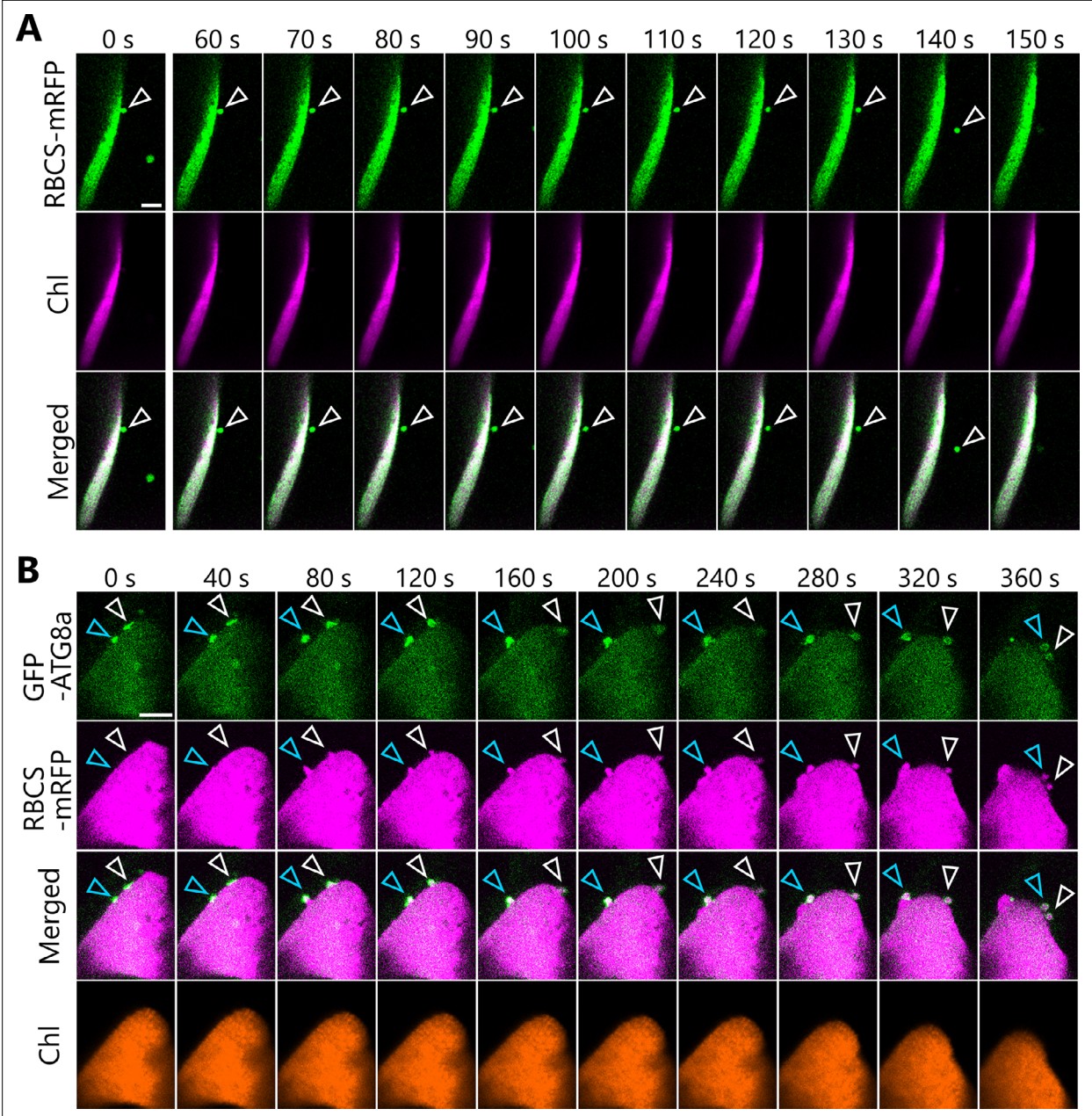

**Figure 9.** Formation and segmentation of chloroplast buds in leaves of the *drp5b* mutant. (**A**) A leaf from the *drp5b* mutant accumulating the stroma marker RBCS-mRFP was incubated in sugar-free solution in darkness for 5 hr from dawn and then observed. Arrowheads indicate a chloroplast bud. Green, RBCS-mRFP; magenta, chlorophyll fluorescence (Chl). (**B**) A leaf from the *drp5b* mutant accumulating the stroma marker RBCS-mRFP and the isolation membrane marker GFP-ATG8a were incubated in sugar-free solution in darkness for 6 hr from dawn and then observed. Second rosette leaves from 21- to 22-day-old plants were used. Arrowheads indicate the position of the chloroplast-associated isolation membrane. White and blue arrowheads indicate different isolation membrane−associated sites in a chloroplast. Images in (**A**) and (**B**) are still frames from *Videos 19 and 20*, respectively. Time scales above the images indicate the elapsed time from the start of the respective videos. Green, GFP-ATG8a; magenta, RBCS-mRFP; orange, chlorophyll (Chl) fluorescence. Scale bars, 5 μm.

*Figure 8 continued*

**Source data 2.** PDF file containing original western blots for *Figure 8C*, indicating the relevant bands, genotypes, and conditions.

**Source data 3.** Source data for the graphs in *Figure 8*.

**Figure supplement 1.** Production of Rubisco-containing bodies (RCBs) in *drp5b* mutants is ATG5 dependent.

**Figure supplement 1—source data 1.** Source data for the graph in *Figure 8—figure supplement 1*.

**Figure supplement 2.** Vacuolar accumulation of stromal marker proteins in sugar-starved leaves.

**Figure supplement 2—source data 1.** Source data for the graph in *Figure 8—figure supplement 2*.

degraded in mutants of chloroplast division ring components might provide insight into the interplay between entire organelle–type autophagy and piecemeal-type autophagy during chloroplast degradation.

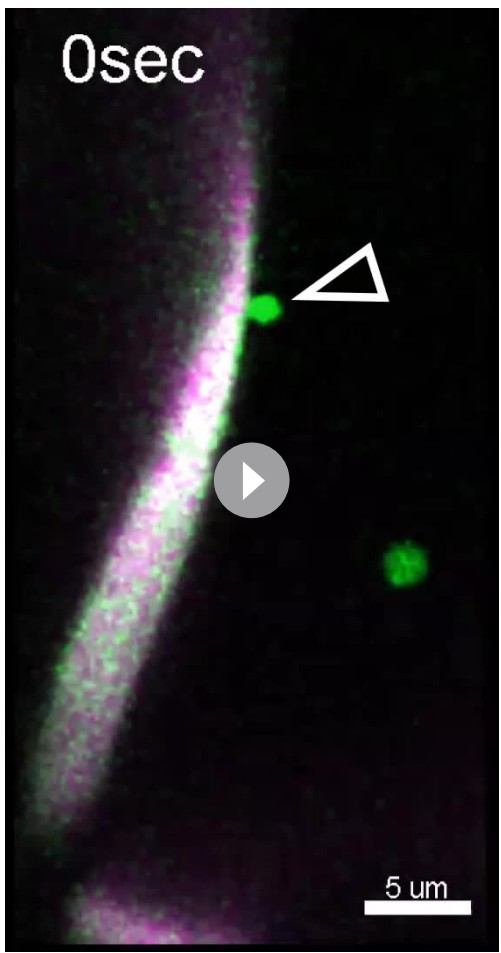

**Video 19.** Release of a chloroplast bud in a sugar-starved leaf of the *drp5b* mutant. The second rosette leaf from a 21-day-old *drp5b* plant accumulating the chloroplast stroma marker RBCS-mRFP was incubated in sugar-free solution in darkness for time-lapse imaging. Arrowhead indicates a chloroplast bud. Images acquired every 1 s are displayed at 20 frames/s. Scale bar, 5 µm. Green, RBCS-mRFP; magenta, chlorophyll fluorescence. Only the video of the merged channels is shown. This video was used to generate *Figure 9A*.

https://elifesciences.org/articles/93232/figures#video19

## Autophagy receptor and adapter proteins for the degradation of mature chloroplasts

Organelle-selective autophagy is typically controlled by receptor proteins that recognize the target organelles and act as a bridge between the organelles and isolation membrane–anchored ATG8 (*Farré and Subramani, 2016*). The observation of a chloroplast-associated isolation membrane (*Figure 6*) suggests that the receptors for chloroplast autophagy connect the isolation membrane to the chloroplast envelope. Such interaction might facilitate the localization of the isolation membrane on the region of subsequent

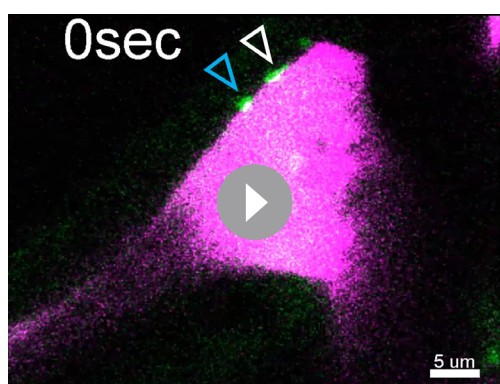

**Video 20.** An enlarged chloroplast caused by the *drp5b* mutation forms Rubisco-containing bodies (RCBs) along the isolation membrane–associated sites. The second rosette leaf from a 22-day-old *drp5b* plant accumulating the chloroplast stroma marker RBCS-mRFP and the isolation membrane marker GFP-ATG8a was incubated in sugar-free solution in darkness for time-lapse imaging. White or blue arrowheads indicate different isolation membrane–associated sites in a chloroplast. Images acquired every 2 s are displayed at 20 frames/s. Scale bar, 5 µm. Green, GFP-ATG8a; magenta, RBCS-mRFP. Only the video of the merged channels is shown. This video was used to generate *Figure 9B*.

https://elifesciences.org/articles/93232/figures#video20

chloroplast budding and its development along the chloroplast surface. In budding yeast, the accumulation of the mitophagy receptor Atg32 is required for the formation of the mitophagosome. Atg32 binds to the mitochondrial outer membrane and interacts with Atg8 and Atg11, the latter being a scaffold protein recruiting other ATG members, for autophagosome formation. In fission yeast (*Schizosaccharomyces pombe*), Atg43 located on the mitochondrion outer membrane binds to Atg8 to stabilize the autophagosomal membrane for mitophagosome formation during starvation (*Fukuda et al., 2020*). In mammalian mitophagy, five types of mitophagy receptors have been identified that bind to LC3 (light chain 3) or GABARAP proteins, the mammalian orthologs of ATG8 (*Onishi et al., 2021*). Whether similar receptors work for chloroplast autophagy in mature leaves has not been established. Atg39 is the receptor for nucleophagy in budding yeast (*Mochida et al., 2015*), during which the condensation of Atg39 enables the protrusion of the nuclear membrane to be sequestered by the autophagosome (*Mochida et al., 2022*). The accumulation of receptor proteins might contribute to chloroplast budding to form RCBs.

Another important regulator of the selective autophagy of organelles is the ubiquitination of the targets of this process. One well-characterized process is mammalian mitophagy regulated by PTEN-induced kinase 1 (PINK1) and Parkin (*Onishi et al., 2021*). PINK1 accumulates on the surfaces of damaged mitochondria, facilitating the ubiquitination of the mitochondria by the E3 ubiquitin ligase Parkin. Subsequently, autophagy adapter proteins connect the ubiquitin chain to LC3/GABARAP proteins for the selective elimination of the ubiquitinated mitochondria via autophagosomes (*Lazarou et al., 2015*). Five autophagy adaptor proteins have been identified in mammals: p62/Sequestosome-1 (SQSTM1), Next to BRCA1 gene 1 (NBR1), Tax1-binding protein 1 (TAX1BP1), nuclear dot protein 52 (NDP52), and optineurin (OPTN). Among these, a plant ortholog of NBR1 has been identified (*Svenning et al., 2011*). A recent study found that *Arabidopsis* NBR1 accumulates on damaged chloroplasts to induce their removal in their entirety (*Lee et al., 2023*). NBR1 may participate in the recognition of damaged chloroplasts during the entire organelle–type degradation of chloroplasts by chlorophagy. Another study demonstrated that NBR1 regulates the selective degradation of the outer envelope–bound TOC protein complexes that control cytoplasm-to-plastid protein import (*Wan et al., 2023*). TOC proteins are ubiquitinated under stress conditions such as ultraviolet-B exposure or high temperature and are then degraded by autophagosomes through NBR1-mediated recognition. These findings suggest that NBR1 plays an important role in the degradation of damaged chloroplast proteins. Whether ubiquitination and NBR1 participate in the degradation of portions of chloroplasts via RCBs under sugar-starvation conditions has not been evaluated.

## Roles of stromules

The molecular functions of stromules are not completely understood. We predicted a functional link between stromules and RCBs such that stromule-associated structures are vesiculated and become RCBs for autophagic degradation. However, our imaging assays revealed that chloroplast stromule formation is not essential for RCB formation in mesophyll cells (*Figure 6*). The use of the *sid2* mutant and NahG transgenic lines indicated the close association between enhanced SA signaling and the elevated stromule formation observed in *atg* mutants (*Figure 7*). This finding is consistent with the suggested role of stromules in inducing programmed cell death as an SA-dependent immune response (*Caplan et al., 2015*). We attribute the enhanced stromule formation in *atg* mutant leaves to the higher SA and ROS levels, alone or in combination with the excess stromal fraction accumulating in these mutants due to impaired RCB formation.

In root cells, an association of the isolation membrane with stromules of non-green plastids was reported (*Spitzer et al., 2015*). Stromule formation is more common in the non-green plastids of tissues without photosynthetic activity than in the mature chloroplasts in leaf mesophyll cells (*Ishida and Yoshimoto, 2008*; *Hanson and Sattarzadeh, 2008*). Therefore, the interaction between stromules and the isolation membrane might be important for the efficient degradation of non-green plastids by autophagy. Our study focused on the degradation mechanism of mature chloroplasts in leaves, since mature chloroplasts are rich in nutrients and amino acids and their degradation is particularly important for plant nutrient recycling (*Makino and Osmond, 1991*). There might also be differences in the degradation mechanism of plastids among cell types.

## Intracellular dynamics for the transport of RCBs

Soluble *N*-ethylmaleimide-sensitive factor attachment protein receptors (SNAREs) mediate versatile membrane fusion events (*Ito and Uemura, 2022*). Multiple SNARE proteins participate in the fusion of the lysosomal/vacuolar membrane to the autophagosomal membrane in mammals and budding yeast. A recent study reported the involvement of the *Arabidopsis* SNARE proteins, VESICLE-ASSOCIATED MEMBRANE PROTEIN 724 (VAMP724) and VAMP726, in autophagosome formation (*He et al., 2023*). In this study, we observed an interesting morphology of vacuolar membranes that incorporate RCBs into the vacuole (*Figure 4*). A similar phenomenon was previously observed during the transport of pexophagosomes (autophagosomes containing peroxisome components) in *Arabidopsis* leaves (*Oikawa et al., 2022*). Thus, the transient engulfment of an autophagosome by the vacuolar membrane might be a common phenomenon for vacuolar incorporation in leaf meso-phyll cells. However, the underlying mechanisms are unknown. In mammals, two autophagosomal membrane–anchored SNAREs, Syntaxin 17 (Stx17) and Ykt6, independently contribute to autophagosome–lysosome fusion (*Itakura et al., 2012*; *Matsui et al., 2018*). Stx17 translocates to the closed autophagosome transiently, enabling its fusion with the lysosome (*Tsuboyama et al., 2016*). Such a translocation system might allow the recruitment of the vacuolar membrane for its fusion with the mature autophagosomes containing RCBs in plant cells.

## Concluding remarks

The current study revealed how a type of piecemeal autophagy transports chloroplast stroma and envelope components into the vacuole for degradation in mature *Arabidopsis* leaves. The key event is the development of the isolation membrane along the chloroplast surface, which may lead to the budding and segmentation of the membrane-contact site of a chloroplast. Proteins mediating chloroplast segmentation and the interaction between the chloroplast surface and the isolation membrane remain to be uncovered. This study developed a live-cell tracking method for piecemeal-type chloroplast autophagy, which will help future studies elucidate the underlying mechanisms of the intracellular dynamics of this type of autophagy.

# Materials and methods

## Plant materials

*Arabidopsis* (*A. thaliana*) plants from the Columbia accession (Col) were used in this study. Plants were grown in soil in growth chambers at 23°C under a 12-hr-light/12-hr-dark photoperiod with illumination from fluorescent lamps or LEDs (90–130 μmol m$^{-2}$ s$^{-1}$). The T-DNA-insertion mutants for *ATG5* (*atg5-1*; SAIL_129_B07), *ATG7* (*atg7-2*; GABI_655B06), *ATG2* (*atg2-1*; SALK_076727), *ATG10* (*atg10-1*; SALK_084434), and *DRP5B* (*arc5-2*; SAIL_71_D11) have been described previously (*Doelling et al., 2002*; *Thompson et al., 2005*; *Miyagishima et al., 2006*; *Phillips et al., 2008*; *Yoshimoto et al., 2009*). The *sid2-2* mutant and a transgenic line expressing NahG have been reported previously (*Delaney et al., 1994*; *Yoshimoto et al., 2009*). The transgenic plants expressing a construct encoding chloroplast stroma–targeted GFP (GFP fused to the transit peptide of RECA protein) under the control of the cauliflower mosaic virus (CaMV) 35S promoter (*Pro35S:CT-GFP*; *Köhler et al., 1997*), chloroplast stroma–targeted DsRed (DsRed fused to the transit peptide of RECA protein) from the CaMV 35S promoter (*Pro35S:CT-DsRed*; *Ishida et al., 2021*), RBCS2B-GFP from the *Arabidopsis RBCS2B* promoter (*ProRBCS:RBCS-GFP*; *Ishida et al., 2008*), VHP1-mGFP from the *Arabidopsis VHP1* promoter (*ProVHP1:VHP1-mGFP*; *Segami et al., 2014*), GFP-ATG8a from the *Arabidopsis UBQ10* promoter (*ProUBQ:GFP-ATG8a*; *Nakamura et al., 2021b*), and TOC64-mRFP from the *Arabidopsis TOC64* promoter (*ProTOC64:TOC64-mRFP*; *Kusano et al., 2023*) have been reported in previous studies. Transgenic plants expressing *RBCS2B-EYFP*, *RBCS2B-mRFP*, or *RBCS2B-tagRFP* from the *RBCS2B* promoter (*ProRBCS:RBCS-EYFP*, *ProRBCS:RBCS-mRFP*, and *ProRBCS:RBCS-tagRFP*) were generated as follows. A genomic fragment encompassing the promoter region and the full-length coding region of *RBCS2B* (At5g38420), cloned into the pENTR/D/TOPO vector (*Ishida et al., 2008*), was inserted into the Gateway vectors pGWB540, pGWB553, or pGWB559 (*Nakagawa et al., 2007*) via LR clonase II (Invitrogen) reaction to generate constructs encoding RBCS2B-EYFP, RBCS2B-mRFP, or RBCS2B-tagRFP, respectively.

Transgenic plants expressing *ATPC1-tagRFP* (*ProATPC1:ATPC1-tagRFP*) were produced as follows. A genomic fragment containing the promoter region and full-length coding region of *ATPC1* (At4g04640) was amplified from Col-0 genomic DNA by PCR using PrimeSTAR DNA polymerase (TaKaRa) and the primers ATPC1_F (CACCCATGGAGAGGGCTCGTACCTTAC) and ATPC1_R (AACC TGTGCATTAGCTCCAG), cloned into pENTR/D-TOPO (Invitrogen), and then recombined into the vector pGWB559 via LR reaction. Transgenic plants expressing *KEA1-mRFP* (*ProKEA1:KEA1-mRFP*) were produced as follows. A genomic fragment containing the promoter region and full-length coding region of *KEA1* (At1g01790) was amplified from Col-0 genomic DNA by PCR using the primers KEA1_F (AGGAACCAATTCAGTCGACTCATGATCATAACAAGTCTC) and KEA1_R (AAAGCTGGGTCTAGAT ATCCGATTACGACTGTGCCTCCTTC), cloned into pENTR1A (Invitrogen) using NEBuilder HiFi DNA Assembly master mix (New England Biolabs), and recombined into the vector pGWB553 via LR reaction. The resulting construct was introduced into *Arabidopsis* plants by the floral dip method (*Clough and Bent, 1998*) using Agrobacterium (*Agrobacterium tumefaciens*) strain GV3101. Transgenic plants expressing two types of fluorescent markers and mutant plants expressing fluorescent markers were generated by crossing or additional transformations.

## Live-cell imaging by confocal microscopy

The second rosette leaves of 20- to 24-day-old plants were used for time-lapse imaging of living cells. The leaves were excised, infiltrated with sugar-free solutions, and incubated in darkness for 5–24 hr, followed by confocal microscopy observations. When the dark treatment was started at the end of the night, the incubation time was around 5–9 hr. When the treatment was started during the light period, the incubation time was around 20–24 hr. The sugar-free solutions contained 10 mM MES-NaOH, pH 5.5, alone or with full-strength Murashige and Skoog salts (Shiotani M. S.) or the photosynthesis inhibitor 3-(3,4-dichlorophenyl)-1,1-dimethylurea. The time scales of each still frame represent the time from the start frame in each time-lapse data.

Time-lapse imaging in *Figure 1* was performed as previously described with a two-photon excitation confocal microscope with a spinning-disk unit (*Otomo et al., 2015*). The GFP signal was excited by 920 nm femtosecond light pulses generated by a mode-locked titanium-sapphire laser light source (Mai Tai eHP DeepSee; Spectra Physics). The YFP, RFP, and chlorophyll signals were excited by 1040 nm femtosecond light pulses generated by an ytterbium laser light source (femtoTrain; Spectra Physics). The fluorescent signals were observed under an inverted microscope (IX-71; Olympus) equipped with a spinning-disk scanner with 100-μm-wide pinholes aligned with a Nipkow disk (CSU-MPϕ100; Yokogawa Electric) and a water-immersion lens (UPLSAPO60XW, numerical aperture [NA] = 1.20, Olympus). The fluorescence images were captured by an EM-CCD camera (EM-C2; Qimaging or iXon Ultra 897; Andor Technology) through a bandpass filter for GFP and YFP (BrightLine 528/38; Semrock), RFP (D630/60M; Chroma Technology), or chlorophyll autofluorescence (BrightLine 685/40; Semrock). For the simultaneous detection of YFP and chlorophyll signals, fluorescence was detected through image-splitting optics (W-View Gemini; Hamamatsu Photonics) including a dichroic mirror (FF580-FDi01-25 36; Semrock) and the bandpass filters. Z-scans were performed with a piezo actuator (P-721; PI). The acquired images were processed and analyzed using NIS-Elements C software (Nikon) or Imaris software (Bitplane).

Time-lapse imaging analysis by confocal laser-scanning microscopy was performed with LSM800 (Carl Zeiss), LSM880 (Carl Zeiss), LSM900 (Carl Zeiss), or SP8 (Leica) systems. A water-immersion objective lens (C-Apochromat 40×, NA = 1.2 or LD C-Apochromat 63×, NA = 1.15; Carl Zeiss) or an oil-immersion objective lens (HC PL APO 63×, NA = 1.40, Leica) was used. Fast Airyscan mode was used for the observation displayed in *Figure 4B* with the LSM880 system.

## Quantification of microscopy images

Confocal images used for the quantification of RCB numbers (*Figure 1—figure supplement 1*, *Figures 5 and 8*), chloroplast protrusions (*Figure 5*), stromules (*Figure 7*), or vacuolar RFP intensity (*Figure 8—figure supplement 2*) were acquired with a C2 system (Nikon) equipped with a water-immersion objective lens (CFI Apochromat LWD Lambda S 40XC, NA = 1.15; Nikon). RFP emission was detected at 580–630 nm (bandpass filter RPB580–630; Omega optical) after excitation with a 559.8-nm diode laser; chlorophyll autofluorescence was detected at 660–720 nm following excitation by a 636.5-nm diode laser; the two signals were detected simultaneously. For the simultaneous

detection of GFP and chlorophyll, GFP emission was detected at 500–550 nm (bandpass filter RPB500–550; Omega Optical) following excitation by a 489.6-nm diode laser; chlorophyll autofluorescence was detected at 660–720 nm following excitation by a 489.6-nm diode laser.

For the quantification of RCBs and chloroplast protrusions, the second rosette leaves of 21-day-old plants were excised, infiltrated with 10 mM MES-NaOH, pH 5.5, containing 1 µM concanamycin A (Santa Cruz), and incubated for 1 day in darkness at 23°C. The stock solution was 100 µM concanamycin A in dimethyl sulfoxide. Sucrose (1%, wt/vol) or full-strength MS salts was added as energy source or nutrients, respectively. The number of accumulated RCBs in a fixed area (215.04 × 215.04 µm each) was counted. The mean number from four areas of one second rosette leaf per plant was calculated from four or five independent plants. The $z$-stack images of the 3D region (215.04 × 215.04 × 15 µm each) were observed, and the proportion of chloroplasts forming protrusions out of 50 chloroplasts was scored in the region. The protruding structures of chloroplast stroma (approximately 0.5–1.5 µm long) containing the chloroplast stroma marker RBCS-mRFP without the chlorophyll signal were defined as chloroplast protrusions in *Figure 5*. The mean from two different regions of one s rosette leaf per plant was calculated from four independent plants.

Chloroplast stromule formation in guard cells was observed in the second rosette leaves of 13-day-old seedlings. The proportion of chloroplasts forming stromules in each pair of guard cells was scored from the $z$-stack images (251.04 × 251.04 × 20 µm each). The mean from ten stomata of one second rosette leaf per plant was calculated from three individual seedlings. Chloroplast stromule formation in mesophyll cells was observed in the third rosette leaves of 20- or 36-day-old plants. The proportion of chloroplasts forming stromules out of 50 chloroplasts was scored in the $z$-stack images (215.04 × 215.04 × 15 µm each). The mean from two different regions of one third rosette leaf per plant was calculated from four individual plants. Thin, extended tubular structures (less than 1 µm in diameter) containing a chloroplast stroma marker (CT-GFP or RBCS-mRFP) without chlorophyll signals were defined as stromules, as described previously (*Hanson and Sattarzadeh, 2011*; *Brunkard et al., 2015*). For the quantification of vacuolar RFP intensity, the second rosette leaves of 21-day-old plants were excised and incubated in 10 mM MES-NaOH, pH 5.5, for 2 days in darkness at 23°C. The RFP intensity in the central area of a mesophyll cell (15.2 × 15.2 µm each) was measured. The mean from twelve cells of one second rosette leaf per plant was calculated from four individual plants.

Confocal images used to evaluate chloroplast stroma volume (*Figure 1—figure supplement 2*), chloroplast buds, and GFP-ATG8a-labeled structures (*Figure 2—figure supplement 1*; *Figure 6—figure supplements 3 and 4*) were acquired with an LSM900 (Carl Zeiss) or SP8 (Leica) system. To quantify chloroplast stroma volume, the second rosette leaves of 21-day-old plants accumulating RBCS-mRFP were excised at dawn, infiltrated with 10 mM MES-NaOH, pH 5.5, and incubated for 24 hr in the dark at 23°C. The $z$-stack images of the 3D region (159.73 × 159.73 × 30 µm each) were acquired, and the volumes of the respective chloroplasts whose entire bodies were captured in the image and could be separated from neighboring chloroplasts as a single structure were measured using Imaris software (Bitplane). Two different regions of one second rosette leaf were observed in five individual plants. Leaves were observed before incubation in the dark as a control. To quantify chloroplast buds, the second rosette leaves of 21- to 23-day-old plants were excised at dawn, infiltrated with 10 mM MES-NaOH, pH 5.5, and incubated for 5–9 hr in the dark at 23°C. The regions with RBCS-GFP, TOC64-mRFP, KEA1-mRFP, ATPC1-tagRFP, or chlorophyll signals within a chloroplast bud and its neighboring chloroplast or chloroplast envelope were selected as the regions of interest, and the fluorescence intensities per unit area were measured in eight individual plants (*Figure 2—figure supplement 1*). The number of chloroplast buds surrounded by GFP-ATG8a signal in four $z$-stack images (159.73 × 159.73 × 20 µm each) of one leaf per plant was scored from five to six individual plants (*Figure 6—figure supplement 3D, E*).

## Protein analysis

The second and third rosette leaves of 21-day-old plants were incubated in darkness in 10 mM MES-NaOH, pH 5.5, for 2 days and then frozen in liquid nitrogen. The leaves from 23-day-old, untreated plants were used as control. The frozen leaves were homogenized using a tissue laser (QIAGEN) and a zirconium bead, before being resuspended in homogenization buffer containing 50 mM HEPES–NaOH, pH 7.5, 16 mM dithiothreitol (DTT), 10% (vol/vol) glycerol, and protease inhibitor cocktail (Nacalai). Following centrifugation at 20,630 × $g$ for 10 min at 4°C, the protein amounts in

the supernatants were measured using a 660-nm Protein Assay Reagent (Pierce). The supernatants were then mixed with an equal volume of sodium dodecyl sulfate (SDS) sample buffer containing 200 mM Tris–HCl, pH 8.5, 20% (vol/vol) glycerol, 2% (wt/vol) SDS, and 0.1 M DTT, and incubated for 5 min at 95°C. An equal amount of protein was subjected to SDS–polyacrylamide gel electrophoresis using TGX FastCast acrylamide gels (Bio-Rad) and transferred to nitrocellulose membrane (Trans-blot turbo transfer pack; Bio-Rad). An anti-RFP 1G9 clone antibody (1:2000, M204-3; MBL) and an anti-cFBPase antibody (1:5000, AS04043; Agrisera) were used as primary antibodies. Goat anti-mouse IgG (H+L) secondary antibody DyLight 800 4X PEG (1:10,000, SA5-35521; Invitrogen) or anti-rabbit HRP secondary antibody (1:10,000, NA934; Cytiba) was used for RFP or cFBPase detection, respectively. The chemiluminescence signals developed with SuperSignal West Dura Extended Duration Substrate (Pierce) and the DyLight 800 fluorescent signals were detected by a ChemiDoc MP system (Bio-Rad). The image processing and the quantification of band intensity were performed using Image Lab Software (Bio-Rad).

## $H_2O_2$ measurements

The amount of $H_2O_2$ in leaf lysates was measured as previously described (*Chakraborty et al., 2016*) with an Amplex Red Hydrogen Peroxide/Peroxidase Assay Kit (Invitrogen) and an Infinite 200 PRO plate reader (Tecan).

## Statistical analysis

Statistical analysis in this study was performed with JMP14.3.0 software (SAS Institute). Student's *t*-test or Tukey's test was used to compare paired samples or multiple samples, respectively.

## Acknowledgements

We thank Emi Sone, Izumi Fukuhara, and Mio Tokuda for their technical support. We thank Dr. Kohki Yoshimoto for the use of *atg*, *sid2*, and NahG plants, Dr. Maureen R Hanson for the use of *Pro35S:CT-GFP* plants, Dr. Shoji Segami for the use of the *ProVHP1:VHP1-mGFP* construct, and Dr. Tsuyoshi Nakagawa for the use of pGWB vectors. We thank the Support Unit for Bio-Material Analysis, RIKEN CBS Research Resources Division, for the use of Leica SP8 system. This work was supported, in part, by KAKENHI (grant numbers JP16H06280, JP18H04852, JP20H04916 to MI, JP20H05352, JP22H04660 to SN, JP22H04627 to HI), the Cooperative Research Program of 'NJRC Mater. & Dev.' (to MI), the Joint Research by Exploratory Research Center on Life and Living Systems (ExCELLS program number 20-314 to MI), and the RIKEN Incentive Research Project (to MI).

## Additional information

### Funding

| Funder | Grant reference number | Author |
| --- | --- | --- |
| Japan Society for the Promotion of Science | JP16H06280 | Masanori Izumi |
| Japan Society for the Promotion of Science | JP18H04852 | Masanori Izumi |
| Japan Society for the Promotion of Science | JP20H04916 | Masanori Izumi |
| Japan Society for the Promotion of Science | JP20H05352 | Sakuya Nakamura |
| Japan Society for the Promotion of Science | JP22H04660 | Sakuya Nakamura |
| Japan Society for the Promotion of Science | JP22H04627 | Hiroyuki Ishida |

| Funder | Grant reference number | Author |
|---|---|---|
| Cooperative Research Program of "NJRC Mater. & Dev." | | Masanori Izumi |
| Joint Research by Exploratory Research Center on Life and Living Systems | 20-314 | Masanori Izumi |
| RIKEN Incentive Research Project | | Masanori Izumi |

The funders had no role in study design, data collection, and interpretation, or the decision to submit the work for publication.

## Author contributions

Masanori Izumi, Conceptualization, Data curation, Formal analysis, Funding acquisition, Validation, Investigation, Visualization, Methodology, Writing - original draft, Project administration, Writing – review and editing; Sakuya Nakamura, Data curation, Formal analysis, Funding acquisition, Investigation, Methodology, Writing – review and editing; Kohei Otomo, Data curation, Methodology, Writing – review and editing; Hiroyuki Ishida, Jun Hidema, Resources, Writing – review and editing; Tomomi Nemoto, Methodology, Writing – review and editing; Shinya Hagihara, Supervision, Writing – review and editing

## Author ORCIDs

Masanori Izumi ![ORCID] https://orcid.org/0000-0001-5222-9163
Sakuya Nakamura ![ORCID] http://orcid.org/0000-0002-8218-6894
Kohei Otomo ![ORCID] http://orcid.org/0000-0002-5322-6295
Hiroyuki Ishida ![ORCID] http://orcid.org/0000-0002-8682-4566
Jun Hidema ![ORCID] http://orcid.org/0000-0002-3798-8257
Tomomi Nemoto ![ORCID] https://orcid.org/0000-0001-6102-1495
Shinya Hagihara ![ORCID] http://orcid.org/0000-0003-0348-7873

Reviewer #1 (Public review): https://doi.org/10.7554/eLife.93232.3.sa1
Reviewer #2 (Public review): https://doi.org/10.7554/eLife.93232.3.sa2
Reviewer #3 (Public review): https://doi.org/10.7554/eLife.93232.3.sa3
Author response https://doi.org/10.7554/eLife.93232.3.sa4

# Additional files

## Supplementary files
• MDAR checklist

## Data availability

All data generated or analyzed during this study are included in the manuscript and supporting files; source data files have been provided for *Figures 1–3 and 5–8*.

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

# Appendix 1

## Appendix 1—key resources table

| Reagent type (species) or resource | Designation | Source or reference | Identifiers | Additional information |
|---|---|---|---|---|
| Gene (*Arabidopsis thaliana*) | RBCS2B | TAIR | AT5G38420 | |
| Gene (*A. thaliana*) | VHP1 | TAIR | AT1G15690 | |
| Gene (*A. thaliana*) | TOC64-III | TAIR | AT3G17970 | |
| Gene (*A. thaliana*) | ATG8a | TAIR | AT4G21980 | |
| Gene (*A. thaliana*) | ATPC1 | TAIR | AT4G04640 | |
| Gene (*A. thaliana*) | KEA1 | TAIR | AT1G01790 | |
| Genetic reagent (*A. thaliana*) | atg5-1 | ABRC | SAIL_129_B07 | |
| Genetic reagent (*A. thaliana*) | atg7-2 | ABRC | GABI_655B06 | |
| Genetic reagent (*A. thaliana*) | atg2-1 | ABRC | SALK_076727 | |
| Genetic reagent (*A. thaliana*) | atg10-1 | ABRC | SALK_084434 | |
| Genetic reagent (*A. thaliana*) | drp5b (arc5-2) | ABRC | SAIL_71_D11 | |
| Genetic reagent (*A. thaliana*) | sid2-2 | ABRC | CS16438 | |
| Genetic reagent (*A. thaliana*) | NahG atg5-1 | 10.1105/tpc.109.068635 | | |
| Genetic reagent (*A. thaliana*) | sid2-2 atg5-1 | 10.1105/tpc.109.068635 | | |
| Genetic reagent (*A. thaliana*) | Pro35S:CT-GFP | 10.1126/science.276.5321.2039 | | |
| Genetic reagent (*A. thaliana*) | Pro35S:CT-DsRed | 10.1093/pcp/pcab084 | | |
| Genetic reagent (*A. thaliana*) | ProRBCS:RBCS-GFP | 10.1104/pp.108.122770 | | |
| Genetic reagent (*A. thaliana*) | ProVHP1:VHP1-mGFP | 10.1105/tpc.114.127571 | | |
| Genetic reagent (*A. thaliana*) | ProUBQ:GFP-ATG8a | 10.1093/pcp/pcaa162 | | |
| Genetic reagent (*A. thaliana*) | ProTOC64:TOC64-mRFP | 10.26434/chemrxiv-2023-kx6gp | | |
| Genetic reagent (*A. thaliana*) | ProRBCS:RBCS-EYFP | This paper | | See 'Plant materials' in Materials and methods |
| Genetic reagent (*A. thaliana*) | ProRBCS:RBCS-mRFP | This paper | | See 'Plant materials' in Materials and methods |
| Genetic reagent (*A. thaliana*) | ProRBCS:RBCS-tagRFP | This paper | | See 'Plant materials' in Materials and methods |
| Genetic reagent (*A. thaliana*) | ProATPC1:ATPC1-tagRFP | This paper | | See 'Plant materials' in Materials and methods |
| Genetic reagent (*A. thaliana*) | ProKEA1:KEA1-mRFP | This paper | | See 'Plant materials' in Materials and methods |
| Antibody | Anti-RFP (Mouse, monoclonal) | MBL | M204-3 | (1:2000) |
| Antibody | Anti-cFBPase (Rabbit, polyclonal) | Agrisera | AS04043 | (1:5000) |
| Recombinant DNA reagent | ProRBCS:RBCS-EYFP | This paper | | See 'Plant materials' in Materials and methods |
| Recombinant DNA reagent | ProRBCS:RBCS-mRFP | This paper | | See 'Plant materials' in Materials and methods |
| Recombinant DNA reagent | ProRBCS:RBCS-tagRFP | This paper | | See 'Plant materials' in Materials and methods |
| Recombinant DNA reagent | ProATPC1:ATPC1-tagRFP | This paper | | See 'Plant materials' in Materials and methods |
| Recombinant DNA reagent | ProKEA1:KEA1-mRFP | This paper | | See 'Plant materials' in Materials and methods |
| Sequence-based reagent | ATPC1_F | This paper | PCR primers (cloning) | CACCCATGGAGAGGGCTCGTACCTTAC |
| Sequence-based reagent | ATPC1_R | This paper | PCR primers (cloning) | AACCTGTGCATTAGCTCCAG |
| Sequence-based reagent | KEA1_F | This paper | PCR primers (cloning) | AGGAACCAATTCAGTCGACTCATGATCATAACAAGTCTC |
| Sequence-based reagent | KEA1_R | This paper | PCR primers (cloning) | AAAGCTGGGTCTAGATATCCGATTACGACTGTGCCTCCTTC |
| Commercial assay or kit | Amplex Red Hydrogen Peroxide/Peroxidase Assay Kit | Invitrogen | A22188 | |
| Chemical compound, drug | Concanamycin A | Santa Cruz | sc-202111 | |

*Appendix 1 Continued on next page*

*Appendix 1 Continued*

| Reagent type (species) or resource | Designation | Source or reference | Identifiers | Additional information |
|---|---|---|---|---|
| Software, algorithm | ZEN | Carl Zeiss | RRID:SCR_013672 | Image processing and quantification (microscopy) |
| Software, algorithm | NIS-Elements C | Nikon | RRID:SCR_020318 | Image processing and quantification (microscopy) |
| Software, algorithm | LAS X | Leica | RRID:SCR_013673 | Image processing and quantification (microscopy) |
| Software, algorithm | Imaris | Bitplane | RRID:SCR_007370 | Image processing and quantification (microscopy) |
| Software, algorithm | Image lab | Bio-Rad | RRID:SCR_014210 | Image processing and quantification (western blot) |
| Software, algorithm | JMP14 | SAS | RRID:SCR_022199 | Statistics |

