## [Editor Report · eLife assessment]

This manuscript investigates how chloroplasts are broken down during light-limiting conditions as plants reorganize their energy-producing organelles during carbon limitation. The authors provide **compelling** live-cell imaging data of plastids and **solid** quantification of events, documenting that buds form on the surface of chloroplasts and pinch away, then associate with the vacuole via a mechanism that depends on autophagy machinery, but not plastid division machinery. This manuscript provides **valuable** groundwork for other scientists studying the regulation and breakdown of energy-producing organelles, including chloroplasts and mitochondria.

---

## [Referee Report · Reviewer #1 (Public review)]

Summary:

The authors demonstrated that carbon depletion triggers the autophagy-dependent formation of Rubisco Containing Bodies, which contain chloroplast stroma material, but exclude thylakoids. The authors show that RCBs bud directly from the main body of chloroplasts rather than from stromules and that their formation is not dependent on the chloroplast fission factor DRP5. The authors also observed a transient engulfment of the RBCs by the tonoplast during delivery to the vacuolar lumen.

Strengths:

The authors demonstrate that autophagy-related protein 8 (ATG8) co-localizes to the chloroplast demarking the place for RCB budding. The authors provide good-quality time-lapse images and co-localization of the markers corroborating previous observations that RCBs contain only stroma material and do not include thylakoid. The text is very well written and easy to follow.

Weaknesses:

The study adds more valuable descriptive information about the previously published phenomenon of RCB formation under carbon starvation but does not reveal the putative mechanisms governing formation of RCBs and their release to the vacuole.

Comments on revised version:

The authors have done an impressive job revising the manuscript and addressed my comments. The authors clarified previous ambiguities and the new version of the manuscript greatly benefits from the provided quantifications and adjusted discussion.

---

## [Referee Report · Reviewer #2 (Public review)]

This manuscript proposed a new link between the formation of chloroplast budding vesicles (Rubisco-containing bodies [RCBs]) and the development of chloroplast-associated autophagosomes. The authors' previous work demonstrated two types of autophagy pathways involved in chloroplast degradation, including piecemeal degradation of partial chloroplast and whole chloroplast degradation. However, the mechanisms underlying piecemeal degradation are largely unknown, particularly regarding the initiation and release of the budding structures. Here, the authors investigated the progression of piecemeal-type chloroplast trafficking by visualizing it with a high-resolution time-lapse microscope. They provide evidence that autophagosome formation is required for the initiation of chloroplast budding, and that stromule formation is not correlated with this process. In addition, the authors also demonstrated that the release of chloroplast-associated autophagosome is independent of a chloroplast division factor, DRP5b.

Overall, the findings are interesting, and in general, the experiments are very well executed.

Comments on revised version:

The authors have generally addressed all of my concerns (and the other reviewer's) and adapted the manuscript where necessary. The revised version has significantly improved the manuscript. From my perspective there are no further concerns.

---

## [Referee Report · Reviewer #3 (Public review)]

Summary:

Regulated chloroplast breakdown allows plants to modulate these energy-producing organelles, for example during leaf aging, or during changing light conditions. This manuscript investigates how chloroplasts are broken down during light-limiting conditions.

The authors present very nice time lapse imaging of multiple proteins as buds form on the surface of chloroplasts and pinch away, then associate with the vacuole. They use mutant analysis and autophagy markers to demonstrate that this process requires the ATG machinery, but not dynamin-related proteins that are required for chloroplast division. The manuscript concludes with discussion of an internally-consistent model that summarizes the results.

Strengths:

The main strength of the manuscript is the high-quality microscopy data. The authors use multiple markers and high-resolution timelapse imaging to track chloroplast dynamics under light limiting conditions.

Weaknesses:

The main weakness of the manuscript is the limited quantitative data. While it can be challenging to quantify dynamic intracellular events, quantification of these processes is important to appreciate the significance of these findings.

---

## [Author Response]

The following is the authors’ response to the original reviews.

**Reviewer #1 (Public Review):**
Summary:The authors demonstrated that carbon depletion triggers the autophagy-dependent formation of Rubisco Containing Bodies, which contain chloroplast stroma material, but exclude thylakoids. The authors show that RCBs bud directly from the main body of chloroplasts rather than from stromules and that their formation is not dependent on the chloroplast fission factor DRP5. The authors also observed a transient engulfment of the RBCs by the tonoplast during delivery to the vacuolar lumen.Strengths:The authors demonstrate that autophagy-related protein 8 (ATG8) co-localizes to the chloroplast demarking the place for RCB budding. The authors provide good-quality time-lapse images and co-localization of the markers corroborating previous observations that RCBs contain only stroma material and do not include thylakoid. The text is very well written and easy to follow.Weaknesses:A significant portion of the results presented in the study comes across as a corroboration of the previous findings made under different stress conditions: autophagy-dependent formation of RCBs was reported by Ishida et all in 2009. Furthermore, some included results are not of particular relevance to the study's aim. For example, it is unclear what is the importance of the role of SA in the formation of stromules, which do not serve as an origin for the RCBs. Similarly, the significance of the transient engulfment of RCBs by the tonoplast remained elusive. Although it is indeed a curious observation, previously reported for peroxisomes, its presentation should include an adequate discussion maybe suggesting the involved mechanism. Finally, some conclusions are not fully supported by the data: the suggested timing of events poorly aligns between and even within experiments mostly due to high variation and low number of replicates. Most importantly, the discussion does not place the findings of this study into the context of current knowledge on chlorophagy and does not propose the significance of the piece-meal vs complete organelle sequestration into the vacuole under used conditions, and does not dwell on the early localization of ATG8 to the future budding place on the chloroplast.

We performed additional experiments with biological replicates that involved quantification. The results of these experiments validate the findings of this study. We also revised the Discussion section, which now includes a discussion of the interplay between piecemeal-type and entire-organelle-type chloroplast autophagy and the relevance of autophagy adaptor and receptor proteins to the localization of ATG8 on the chloroplast surface. Accordingly, the first subheading section in the Discussion became too long. Therefore, we divided it into two subheading sections. We believe that the revisions successfully address the weaknesses pointed out by the reviewer and enhance the importance of the current study. Below is a detailed description of the improvements made to our manuscript in response to the reviewer comments.

**Reviewer #1 (Recommendations For The Authors):**
It would be great if the authors kindly used numbered lines to facilitate the review process.

We have added line numbers to the text of the revised version of the manuscript.

The authors use the words "budding", "protrusion" and "stromule formation" interchangeably in some parts of the text. For the sake of clarity, it would be best to be consistent in the terminology and possibly elaborate on the exact differences between these structure types and the criteria by which they were identified.

We have checked all of the text and improved the consistency of the terminology. An important finding of this study is that chloroplasts form budding structures at the site associated with ATG8. These structures then divide to become a type of autophagic cargo termed a Rubiscocontaining body. We therefore mainly use the terms “bud” and “budding” throughout the text. In the experiments shown in Figure 5, we considered the possibility that chloroplast protrusions accumulate in leaves of *atg* mutants and do not divide because the mutants cannot create autophagosomes. Therefore, the word “protrusion” was used to describe the results shown in Figure 5 in which the proportion of chloroplasts forming protrusions was scored. In the revised text, the word “protrusion” is only used in descriptions of Figure 5. Previous reports define stromules as thin, tubular, extended structures (less than 1 µm in diameter) of the plastid stroma (Hanson and Sattarzadeh, 2011; Brunkard et al., 2015). In the revised text, the word “stromules” is used to describe the structures defined in these previous reports. We have added definitions of each term to the Introduction, Methods and Results sections where appropriate (lines 57–58, 160–162, 247–249, 313–316, 655–658, 668–670).

Pages 3-4: the authors observed budding of the chloroplasts within a few minutes - it would be helpful to specify that time was probably counted from the first observation of budding, not from the start of the dark treatment, and also specify the exact treatment duration for each of the experiments.

The time scales in the figures do not represent the time from the start of the dark treatment. Instead, they describe the duration from the start of the time-lapse videos that were used to generate the still images. Therefore, the indicated time scales are almost the same as the duration from the start of the observations of each target structure (chloroplast buds or GFPATG8a-labeled structures). As described in the Methods section, leaves were incubated in darkness for 5 to 24 h to induce sugar starvation. Such sugar-starved leaves were subjected to live-cell monitoring for the target structures. Since Arabidopsis leaves accumulate starch as a stored sugar source (Smith and Stitt, 2007; Usadel et al., 2008), dark treatment lasting several minutes is not sufficient for the starch to be consumed and sugar starvation to be induced. To avoid confusion, we have added definitions of the time scales to the legends of figures containing the results of time-lapse imaging. We have also specified the durations of dark treatments used to obtain the respective results in the legends.

Figure 6: the time scale for complete autophagosome formation is in the range of 100-120 sec, how do these results align with the results shown in Figures 3B and C, where complete autophagosomes are suggested to be released into the vacuole after 73.8 sec. Furthermore, another structure is suggested to be formed within 50 sec. Such experiments possibly require a large number of replicates to estimate representative timing.

As mentioned in the previous response, the time scales in still frames represent the duration from the start of the corresponding video. Leaves incubated in darkness for 5 to 24 h were subjected to live-cell imaging. When we identified the target structures, e.g., GFP-ATG8alabeled structures on the surfaces of chloroplasts (Figure 6) or chloroplast budding structures (Figure 3), we began to track these structures. Therefore, the time scales in the figures do not align to a common time axis. We revised the descriptions about Figure 3 and Figure 6 in the Results section to clearly explain that the time points in each experiment merely indicates the time of one observation.

The authors might want to consider using arrows to indicate structures of interest in all movies and figures.

We have added arrows to indicate the structures of interest in the starting frames of all videos. We hesitate to add arrows to highlight RCBs accumulating in the vacuole (Figure 1-figure supplement 1, Figure 5 and Figure 8) and stromules (Figure 7) because many arrows would be required, which would obscure large portions of the images. We believe that the images without arrows clearly represent the appearance of RCBs or stromules and that their quantification (Figure 1-figure supplement 1C, Figure 5B, Figure 5-figure supplement 1B, Figure 7B, 7D, 7F, and Figure 8B) well supports the results.

Figure 7 Supplement 1: do the authors detect complete chloroplasts in the vacuole of atg7 and sid2/atg7?

We did not observe the vacuolar transport of whole chloroplasts in *atg7* or *atg7 sid2* plants under our experimental conditions. The figure below (Figure 1 for Response to reviewers) shows images of mesophyll cells from a leaf (third rosette leaf of a 20-d-old plant) of *atg7* accumulating chloroplast stroma–targeted GFP (CT-GFP); this is from the previous version of Figure 7–figure supplement 1. Indeed, some GFP bodies exhibiting strong stromal GFP (CTGFP) signals appeared in the central area of the cell (arrowheads in A). However, such bodies were chloroplasts in epidermal cells. The 3D images (B) and cross-section image (x to z axis) of the region highlighted by the blue dotted line (C) indicate that such GFP bodies are the edges of chloroplasts that localize on the abaxial side of the observed region. Because CT-GFP expression was driven by the 35S promoter, strong GFP signals appeared in chloroplasts in epidermal cells in addition to chloroplasts in mesophyll cells. Previous studies using the same transgenic lines also showed that chloroplasts in epidermal cells exhibit strong GFP signals (Kohler et al., 1997; Caplan et al., 2015; Lee et al., 2023). RBCS-mRFP or GFP driven by the RBCS2B promoter do not label the chloroplasts in epidermal cells (new Figure 7-figure supplement 1). Additionally, because the borders between the mesophyll cell layer and the epidermal cell layer are not even, chloroplasts in epidermal cells are sometimes visible during observations of mesophyll cells. Such detection more frequently occurs during the acquisition of *z*-stack images. This point was more precisely demonstrated in our previous study with the aid of Calcofluor white staining of cell walls (Nakamura et al., 2018). Please see Supplemental Figure S3 in our previous report. To avoid any misunderstanding, we replaced the image of the leaf from *atg7* in the revised figure, which is now Figure 7-figure supplement 2, with an image of another region to more precisely visualize mesophyll cells in this plant line.

**Author response image 1. sa4fig1:** Mesophyll cells in a leaf of *atg7* accumulating stromal CT-GFP, reconstructed from the data shown in the previous version of Figure 7–figure supplement 1. (A) Individual channel images (CT-GFP and chlorophyll) from the merged orthogonal projection image shown in the previous version of Figure 7–figure supplement 1. The right panel shows the enhanced chlorophyll signal to clearly visualize the chloroplasts in epidermal cells. Green, CTGFP; magenta, chlorophyll fluorescence. Scale bar, 20 µm. (B) 3D structure of the merged image shown in (A). (C) Images of the cross section indicated by the blue dotted line (a to b) in B. Arrowheads indicate the edges of chloroplasts in epidermal cells.

Figure 8: it would be interesting to hear the authors' opinion on why they observed a significant increase in RCBs number in the drp5b mutant background

We have added a discussion of this issue to the revised manuscript (lines 445–459). We now have two hypotheses to explain this issue. One hypothesis is that the impaired chloroplast division due to the *drp5b* mutation reduces energy availability and thus activates chloroplast autophagy. The other hypothesis is that the *drp5b* mutation impairs the type of chlorophagy that degrades whole chloroplasts, and thus piecemeal-type chloroplast autophagy via Rubiscocontaining bodies is activated. However, we do not have any experimental evidence supporting either hypothesis.

**Reviewer #2 (Public Review):**
This manuscript proposed a new link between the formation of chloroplast budding vesicles (Rubisco-containing bodies [RCBs]) and the development of chloroplast-associated autophagosomes. The authors' previous work demonstrated two types of autophagy pathways involved in chloroplast degradation, including piecemeal degradation of partial chloroplast and whole chloroplast degradation. However, the mechanisms underlying piecemeal degradation are largely unknown, particularly regarding the initiation and release of the budding structures. Here, the authors investigated the progression of piecemeal-type chloroplast trafficking by visualizing it with a high-resolution time-lapse microscope. They provide evidence that autophagosome formation is required for the initiation of chloroplast budding, and that stromule formation is not correlated with this process. In addition, the authors also demonstrated that the release of chloroplast-associated autophagosome is independent of a chloroplast division factor, DRP5b.Overall, the findings are interesting, and in general, the experiments are very well executed. Although the mechanism of how Rubisco-containing bodies are processed is still unclear, this study suggests that a novel chloroplast division machinery exists to facilitate chloroplast autophagy, which will be valuable to investigate in the future.
**Reviewer #2 (Recommendations For The Authors):**
Below are some specific comments.(1) In Supplement Figure 1B, there is no chloroplast stromule in RBCS-mRFP x atg7-2 plants under dark treatment with ConA, but in Figure 7A, there are stromules in CT-GFP x atg7-2 plants. How to explain such a discrepancy? Did the authors check the chloroplast morphology of RBCS-mRFP x atg7-2 plants in different developmental stages? Will it behave the same as CT-GFP x atg7-2 under the same condition as in Figure 7A?

As described in the text, the ages and conditions of the leaves shown in Figure 1–figure supplement 1 and Figure 7 are different. In Figure 1–figure supplement 1, second rosette leaves from 21-d-old plants were incubated in the dark with concanamycin A for 1 d. In Figure 7E and 7F, we explored the condition under which mesophyll chloroplasts in *atg* leaves actively form stromules to assess how a deficiency in autophagy is related to stromule formation. We found that late senescing leaves (third rosette leaves from 36-d-old plants) of *atg5* and *atg7* plants accumulated many stromules without additional treatment (Figure 7). It is not surprising that the chloroplast morphologies shown in Figures 1 and 7 are different because the leaf ages and conditions are largely different.

However, we agree that the differences in chloroplast stroma–targeted GFP and RBCS-mRFP might influence the visualization of stromules. For instance, fluorescent protein– labeled RBCS proteins are incorporated into the Rubisco holoenzyme, comprising eight RBCS and eight RBCL proteins (Ishida et al., 2008; Ono et al., 2013). Such a large protein complex might not accumulate in stromules. Therefore, we examined the chloroplast morphology in late senescing leaves (third rosette leaves from 36-d-old plants) from WT, *atg5*, and *atg7* plants harboring *ProRBCS:RBCS-mRFP*, as you suggested. Mesophyll chloroplasts formed many stromules in *atg5* and *atg7* leaves but not in WT leaves (Figure 7–figure supplement 1). These results indicate that RBCS-mRFP can be used to visualize stromules and that the differences in chloroplast morphology between Figure 1-figure supplement 1 and Figure 7 cannot be attributed to the different marker proteins used. A previous study also indicated that Rubisco is present in plastid stromules (Kwok and Hanson, 2004).

(2) In Figure 2, the author showed that the outer envelope marker Toc64 was colocalized with chloroplast buds. How about proteins in the inner envelope membrane of chloroplasts?

We generated Arabidopsis plants expressing red fluorescent protein–tagged K+ EFFLUX ANTIPORTER 1 (KEA1), a chloroplast inner envelope membrane protein (Kunz et al., 2014; Boelter et al., 2020). We found that the chloroplast buds visualized by RBCS-GFP were also marked by KEA1-mRFP (Figure 2–figure supplement 1B). We observed the transport of such buds (Figure 2–figure supplement 2). These results strengthen our claim that autophagy degrades chloroplast stroma and envelope components as a type of specific cargo termed a Rubisco-containing body. The descriptions about this additional experiment are in lines 181– 187.

(3) In Figure 3, how many RCBs were tracked for the trafficking analysis to raise the conclusion that the vesicle was released into the vacuole around 73.8s?

We apologize for our confusing explanation in the previous version of the manuscript. The time point “73.8 s” merely indicates the time of one observation, as shown in Figure 3. This time does not represent the common timing of vacuolar release of a Rubisco-containing body. As we explained in the response to the comments from reviewer 1, we subjected leaves that were incubated in the dark for several hours to live-cell imaging assays to observe chloroplast morphology in sugar-starved leaves. The time scales of each still frame represent the time from the start of the corresponding video. Therefore, the time points in the respective figures do not align to a common time axis, and the number “73.8 s” is not important. We attempted to emphasize that the type of movement of Rubisco-containing bodies changes during their tracking shown in Figure 3. Based on this finding, we hypothesized that the Rubisco-containing bodies are released into the vacuolar lumen when they initiate random movement. Therefore, we expected that the interaction between the Rubisco-containing bodies and the vacuolar membrane could be captured, and we therefore turned our attention to the dynamics of the vacuolar membrane in subsequent experiments. Accordingly, our observations of the vacuolar membrane allowed us to visualize the release of the Rubisco-containing body into the vacuole (Figure 4). We rephrased these sentences (lines 212–219) to avoid confusion and to explain this idea accurately. We also performed tracking experiments of Rubisco-containing bodies to strengthen the finding that the type of movement of the bodies changes during tracking (Figure 3-figure supplement 1, Videos 8 and 9).

(4) I do believe the conclusion that vacuolar membranes incorporate RCBs into the vacuole in Figure 4. However, it will be more convincing if images of higher quality are provided.

We tried to acquire images that more clearly show the morphology of the vacuolar membrane during the incorporation of the Rubisco-containing body. We obtained the images in Figure 4A using a standard type of confocal microscope, the LSM 800 (Carl Zeiss), and obtained the images in Figure 4B using the Airyscan Fast acquisition mode, a hyper-resolution microscope mode, in the LSM 880 system (Carl Zeiss). We performed additional experiments with another type of confocal microscope, the SP8 (Leica; Figure 4-figure supplement 1A to 1C, Videos 12– 14). The quality of the images from these experiments was as high as possible under the experimental conditions (equipment and plant materials). In general, increasing the image resolution during time-lapse imaging with a confocal microscope requires reducing the time resolution. However, the transport of a Rubisco-containing body occurs relatively quickly: Its engulfment by the vacuolar membrane takes place for just a few seconds (Figure 4, Figure 4figure supplement 1). We could therefore not reduce the time resolution further to better capture the morphology of the vacuolar membrane.

(5) In Figure 7G, the authors concluded that SA and ROS might be the cause of the extensive formation of stromules. How about the H2O2 level in NahG and atg5 NahG plants? Compared with sid2, NahG appeared to completely inhibit stromule formation in atg5. Will this be related to ROS levels?

We measured the hydrogen peroxide (H2O2) contents in NahG *atg5* plants and *atg5* single mutant plants and found that their leaves accumulate more H2O2 than those of wild-type or NahG plants (Figure 7-figure supplement 3). Since we have only maintained fresh seeds of NahG *atg5* plants harboring the *35S* promoter–driven chloroplast stroma–targeted GFP (*Pro35S:CT-GFP*) construct, we first confirmed that CT-GFP accumulation does not affect the measurement of H2O2 content. H2O2 levels were similar between wild-type leaves and CT-GFPexpressing leaves. A comparison among *Pro35S:CT-GFP* expressing lines in the wild-type, *atg5*, NahG, and NahG *atg5* backgrounds revealed enhanced accumulation of H2O2 in the *atg5* and NahG *atg5* genotypes compared with the wild-type and NahG genotypes. This finding is consistent with the results of histological staining of H2O2 using 3,3′-diaminobenzidine (DAB) in a previous study (Yoshimoto et al., 2009).

It is unclear why NahG expression inhibited stromule formation more strongly than the *sid2* mutation in the *atg5* mutant background, as you pointed out (Figure 7A–D). NahG catabolizes salicylic acid (SA), whereas *sid2* mutants are knockout mutants of *ISOCHORISMATE SYNTHASE1* (*ICS1*), a gene required for SA biosynthesis. Plants have two metabolic routes for SA biosynthesis: The isochorismate synthase (ICS) pathway and the phenylalanine ammonia-lyase (PAL) pathway. Furthermore, Arabidopsis plants contain two ICS homologs: ICS1 and ICS2. Previous studies have revealed that ICS1 (SID2) is the main player for SA biosynthesis in response to pathogen infection (Delaney et al., 1994). Another study revealed drastically lower SA contents in the leaves of both *sid2* single mutants and NahGexpressing plants compared with those of wild-type plants (Abreu and Munné-Bosch, 2009). Therefore, it is clear that the *sid2* single mutation sufficiently inhibits SA accumulation in Arabidopsis leaves. However, low levels of SA biosynthesis through ICS1-independent routes might influence stromule formation in leaves of *sid2 atg5* and *sid2 atg7*. Because a previous study demonstrated that the *sid2* single mutation sufficiently suppresses the SA hyperaccumulation–related phenotypes of *atg* plants (Yoshimoto et al., 2009), we believe that the use of the *sid2* mutation was adequate to assess the effects of SA on stromule formation that actively occurs in the *atg* plants examined in this study.

(6) In Supplement Figure 7, I have noticed that there are still some CT-GFP signals (green dots) in the vacuoles of the atg7 mutant, are they RCBs? If so, how can this phenomenon be explained?

As we explained in the response to the comment from Reviewer 1, CT-GFP-labeled bodies are chloroplasts in the epidermal cell layer. Please see our response to Reviewer 1’s comment about Figure 7 and the associated figure (Figure 1 for Response to reviewers). The CT-GFP-labeled dots (arrowheads) are the edges of chloroplasts and localize on the abaxial side of the observed region. The dots have faint chlorophyll signals. This phenomenon is much more clear in the image with enhanced brightness (right panel in A). Since the bodies are merely the edges of epidermal chloroplasts, their chlorophyl signals are faint. Therefore, these bodies are not Rubisco-containing bodies but are instead simply the edges of chloroplasts in the epidermal cell layer.

(7) On page 24, the second paragraph, lines 12-14, the authors claim that no receptors similar to those involved in mitophagy that bind to LC3 (ATG8) have been established in chloroplasts. Actually, it has been reported that a homologue of mitophagy receptor, NBR1, acts as an autophagy receptor to regulate chloroplast protein degradation (Lee et al, 2023, Elife; Wan et al, 2023, EMBO Journal). Although I do think NBR1 is not involved in RCBs based on these reports, these findings should be discussed here.

Thank you for this good suggestion. We have added a discussion about this important point to the Discussion section, along with the relevant citations (lines 482–502).

(8) In the figure legend, the details of the experiments will be better provided, such as leaves stages (Figure 1, Figure 5...), the number of chloroplasts analyzed (Figure 7...). This can help the readers to follow.

Thank you for highlighting this. We have checked all of the figure legends and added descriptions of the leaf stages and experimental conditions.

**Reviewer #3 (Public Review):**
Summary:Regulated chloroplast breakdown allows plants to modulate these energy-producing organelles, for example during leaf aging, or during changing light conditions. This manuscript investigates how chloroplasts are broken down during light-limiting conditions.The authors present very nice time-lapse imaging of multiple proteins as buds form on the surface of chloroplasts and pinch away, then associate with the vacuole. They use mutant analysis and autophagy markers to demonstrate that this process requires the ATG machinery, but not dynamin-related proteins that are required for chloroplast division. The manuscript concludes with a discussion of an internally-consistent model that summarizes the results.Strengths:The main strength of the manuscript is the high-quality microscopy data. The authors use multiple markers and high-resolution time-lapse imaging to track chloroplast dynamics under light-limiting conditions.Weaknesses:The main weakness of the manuscript is the lack of quantitative data. Quantification of multiple events is required to support the authors' claims, for example, claims about which parts of the plastid bud, about the dynamics of the events, about the colocalization between ATG8 and the plastid stroma buds, and the dynamics of this association. Without understanding how often these events occur and how frequently events follow the manner observed by the authors (in the 1 or 2 examples presented in each figure) it is difficult to appreciate the significance of these findings.

We have performed several additional experiments, including the quantification of multiple chloroplast buds or GFP-ATG8-labeled structures from individual plants. The results strengthen our claims and thus improve the significance of the current study. Please see the responses below for details.

**Reviewer #3 (Recommendations For The Authors):**
Overall, the live-cell imaging in this paper is high quality and rigorously conducted. However, without quantification of these events, it is difficult to judge whether this is an occasional contributor to plastid breakdown, or the primary mechanism for this process.- For Figure 1, the authors could estimate the importance of this mechanism for chloroplast breakdown by calculating the volume change in chloroplasts over time during light-limiting conditions, then comparing this to the volume of the puncta that bud off of plastids and the frequency of these events. That is, what percentage of chloroplast volume loss can be accounted for by puncta that bud from chloroplasts? Are there likely other mechanisms contributing to chloroplast breakdown, or is this the primary mechanism?

We measured the volumes of chloroplast stroma when the leaves from wild-type (WT) and *atg7* plants accumulating RBCS-mRFP were subjected to extended darkness for 1 d (Figure 1-figure supplement 2). The volume of the chloroplast stroma in dark-treated leaves of WT plants was 70% that in leaves before treatment, whereas the volume of the chloroplast stroma in darktreated *atg7* leaves was 86% that in leaves before treatment. The transport of Rubiscocontaining bodies into the vacuole did not occur in *atg7* leaves (Figure 1-figure supplement 1). These results suggest that the release of chloroplast buds as Rubisco-containing bodies contributes to the decrease in chloroplast stroma volume during dark treatment. These results also suggest that autophagy-independent systems contribute to the decrease in chloroplast volume. It is difficult to monitor the volume or frequency of budding off of puncta from chloroplasts during dark treatment because the budding and transport of the puncta occur relatively quickly and are completed within minutes, and the puncta frequently move away from the plane of focus. Additionally, continuous monitoring of chloroplast morphology over the dark treatment period requires the long-term exposure of leaves to repeated laser excitation, and such treatment might cause unexpected stress. We believe that the evaluation of chloroplast stroma volume after 1 d of dark treatment is important for estimating the contribution of the mechanism described in this study. The descriptions about this additional experiment are in lines 163–174.

- The claim that structures budding from the plastid "specifically contains stroma material...without any chlorophyll signal" (p. 6 and Figure 2) should be supported by quantitative analysis of many such buds in multiple cells from multiple independent plants.

We performed additional experiments (Figure 2-figure supplement 1) to measure the fluorescence intensity ratios of the stroma marker RBCS-GFP and chlorophyll between chloroplast budding structures and their neighboring chloroplasts in Arabidopsis plants expressing the stromal marker RBCS-GFP along with TOC64-mRFP (a chloroplast outer envelope membrane protein), KEA1-mRFP (a chloroplast inner envelope membrane protein), or ATPC1-tagRFP (a thylakoid membrane protein). The results indicated that chloroplast buds contain chloroplast stroma without chlorophyll signals. The descriptions of this experiment are in lines 175–199. In these experiments, we observed 30 to 33 chloroplast buds from eight individual plants.

- Claims about the dynamics of these events in Figures 2 & 3 should be supported by quantitative analysis of many buds in multiple cells from multiple independent plants and appropriate summary statistics (e.g. mean, standard deviation), and claims about the coordination of events should be supported by statistical comparison of these measurements between different markers.

As mentioned in the response to the above comments, quantification of fluorescent intensities (Figure 2-figure supplement 1) revealed that the chloroplast budding structures produced TOC64-mRFP and KEA1-mRFP signals without ATPC1-tagRFP signal. These results support the claim that chloroplast buds contain chloroplast stroma and envelope components without thylakoid membranes.

It is not easy to quantify the dynamics of chloroplast buds since the puncta sometimes move away from the plane of focus. We therefore added data from individual time-lapse observations showing that the type of movement exhibited by the puncta changes during tracking (Figure 3-figure supplement 1A and 1B, Videos 8 and 9) to strengthen the notion that such a phenomenon was observed repeatedly.

- Data in Figure 4 should be supported by quantification of the proportion of plastid-derived puncta that end up inside the vacuole (compared to those that do not) in multiple cells from multiple independent plants.

Although we performed additional observations of the destinations of chloroplast-derived puncta, we encountered some difficulty in correctly calculating the proportion of plastid-derived puncta that ended up inside the vacuole. This problem is similar to the difficulty in tracking Rubisco-containing bodies mentioned in the response to the previous comments. During timelapse imaging, puncta sometimes move from the plane of focus toward the deeper side (abaxial side) or near side (adaxial side), causing us to lose track of a number of puncta. Therefore, we could not determine the destinations of all puncta to calculate the proportion of puncta that ended up in the vacuolar lumen.

Alternatively, we added the results of three experiments (Figure 4-figure supplement 1, Videos 12–14) examining how the vacuolar membrane engulfs the chloroplast-derived puncta to incorporate them inside the vacuole. The data support the notion that such a phenomenon occurs repeatedly in sugar-starved leaves. All results were obtained from individual plants.

- Data in Figure 6 should also be supported by quantitative analysis of many buds in multiple cells from multiple independent plants, to determine whether ATG8 associates with all RBCScontaining buds, and vice versa.

To address this issue, we performed additional experiments on plants expressing GFP-ATG8a and RBCS-mRFP (Figure 6-figure supplements 3 and 4). First, we observed 58 chloroplast buds from eight individual plants and evaluated the proportion of GFP-ATG8a-labeled chloroplast buds. We determined that 64% of chloroplast buds were at least autophagy-associated structures (Figure 6-figure supplement 3A–3C). This result also suggests that chloroplasts can form autophagy-independent budding structures, which might be associated with stromule-related structures or the autophagy-independent vesiculation machinery. We also evaluated the number of GFP-ATG8a-labeled chloroplast buds (Figure 6-figure supplement 3D and 3E). The formation of such structures increased in response to dark treatment (Figure 6-figure supplement 3D), but they did not appear in *atg7* plants exposed to the dark (Figure 6-figure supplement 3E). These results support the notion that the formation of chloroplast buds to be released as Rubisco-containing bodies requires the core ATG machinery.

Furthermore, we observed 157 GFP-ATG8a-labeled structures from thirteen individual plants and evaluated the proportion of chloroplast-associated isolation membranes (Figure 6-figure supplement 4). We also classified the chloroplast-associated, GFP-ATG8alabeled structures into two categories: the chloroplast surface type (Figure 7-figure supplement 4A) and the chloroplast bud type (Figure 7-figure supplement 4B). This experiment suggested that 43% of the isolation membranes labeled by GFP-ATG8a were involved in chloroplast degradation during an early phase of sugar starvation (extended darkness for 5 to 9 h from the end of night) in mesophyll cells. We believe that these results indicate that autophagy contributes substantially to chloroplast degradation via the morphological changes observed in this study. The descriptions about these experiments are in lines 284–300 in the Results section and in lines 426–444 in the Discussion section.

- Which parts of the plastid bud (Fig 2), about the dynamics of the events (Fig 3), about the colocalization between ATG8 and the plastid stroma buds, and the dynamics of this association (Fig 6).

We performed multiple quantitative studies to address the issues listed above. We believe that these additional experiments strengthened our findings.

- I suggest that the authors avoid using the term "vesicles" to describe the plastid-derived puncta, since it doesn't seem like coat proteins are required for their formation. I suggest "puncta" or similar terms.

We replaced the term “vesicles” with “puncta” or other suitable terms, as suggested.

References for response to reviewers

Abreu ME, Munné-Bosch S (2009) Salicylic acid deficiency in transgenic lines and mutants increases seed yield in the annual plant. J Exp Bot 60: 1261-1271.

Boelter B, Mitterreiter MJ, Schwenkert S, Finkemeier I, Kunz HH (2020) The topology of plastid inner envelope potassium cation efflux antiporter KEA1 provides new insights into its regulatory features. Photosynth Res 145: 43-54.

Brunkard JO, Runkel AM, Zambryski PC (2015) Chloroplasts extend stromules independently and in response to internal redox signals. Proc Natl Acad Sci U S A 112: 10044-10049.

Caplan JL, Kumar AS, Park E, Padmanabhan MS, Hoban K, Modla S, Czymmek K, Dinesh-Kumar SP (2015) Chloroplast stromules function during innate immunity. Dev Cell 34: 45-57.

Delaney TP, Uknes S, Vernooij B, Friedrich L, Weymann K, Negrotto D, Gaffney T, Gutrella M, Kessmann H, Ward E, Ryals J (1994) A Central Role of Salicylic-Acid in Plant-Disease Resistance. Science 266: 1247-1250.

Hanson MR, Sattarzadeh A (2011) Stromules: Recent Insights into a Long Neglected Feature of Plastid Morphology and Function. Plant Physiol 155: 1486-1492.

Ishida H, Yoshimoto K, Izumi M, Reisen D, Yano Y, Makino A, Ohsumi Y, Hanson MR, Mae T (2008) Mobilization of rubisco and stroma-localized fluorescent proteins of chloroplasts to the vacuole by an ATG gene-dependent autophagic process. Plant Physiol 148: 142-155.

Kohler RH, Cao J, Zipfel WR, Webb WW, Hanson MR (1997) Exchange of protein molecules through connections between higher plant plastids. Science 276: 2039-2042.

Kunz HH, Gierth M, Herdean A, Satoh-Cruz M, Kramer DM, Spetea C, Schroeder JI (2014) Plastidial transporters KEA1, -2, and -3 are essential for chloroplast osmoregulation, integrity, and pH regulation in. Proc Natl Acad Sci U S A 111: 74807485.

Lee HN, Chacko JV, Solis AG, Chen KE, Barros JA, Signorelli S, Millar AH, Vierstra RD, Eliceiri KW, Otegui MS, Benitez-Alfonso Y (2023) The autophagy receptor NBR1 directs the clearance of photodamaged chloroplasts. Elife 12: e86030.

Ono Y, Wada S, Izumi M, Makino A, Ishida H (2013) Evidence for contribution of autophagy to rubisco degradation during leaf senescence in *Arabidopsis thaliana*. Plant Cell Environ 36: 1147-1159.

Smith AM, Stitt M (2007) Coordination of carbon supply and plant growth. Plant Cell Environ 30: 1126-1149.

Usadel B, Blasing OE, Gibon Y, Retzlaff K, Hoehne M, Gunther M, Stitt M (2008) Global transcript levels respond to small changes of the carbon status during progressive exhaustion of carbohydrates in Arabidopsis rosettes. Plant Physiol 146: 1834-1861.

Yoshimoto K, Jikumaru Y, Kamiya Y, Kusano M, Consonni C, Panstruga R, Ohsumi Y, Shirasu K (2009) Autophagy negatively regulates cell death by controlling NPR1dependent salicylic acid signaling during senescence and the innate immune response in *Arabidopsis*. Plant Cell 21: 2914-2927.